# BEYOND EDGE DELETION: A COMPREHENSIVE APPROACH TO COUNTERFACTUAL EXPLANATION IN GRAPH NEURAL NETWORKS

## ABSTRACT

Graph Neural Networks (GNNs) are increasingly adopted in domains like molecular biology and social network analysis, yet their black-box nature hinders interpretability and trust. This is especially problematic in high-stakes applications, such as predicting molecule toxicity, drug discovery, or guiding financial fraud detections, where transparent explanations are essential. Counterfactual explanations – minimal changes that flip a model's prediction – offer a transparent lens into GNNs' behavior. In this work, we introduce XPlore, a novel technique that significantly broadens the counterfactual search space. It consists of gradient-guided perturbations to adjacency and node feature matrices. Unlike most prior methods, which focus solely on edge deletions, our approach belongs to the growing class of techniques that optimize edge insertions and node-feature perturbations, here jointly performed under a unified gradient-based framework, enabling a richer and more nuanced exploration of counterfactuals. To quantify both structural and semantic fidelity, we introduce a cosine similarity metric on learned graph embeddings, addressing a key limitation of traditional distance-based metrics, demonstrating that XPlore produces more coherent and minimal counterfactuals. Empirical results on 13 real-world and 5 synthetic benchmarks show up to +56.3% improvement in validity and +52.8% in fidelity over state-of-the-art baselines, while retaining competitive runtime.

## 1 INTRODUCTION

Explainability is crucial in fields such as healthcare and finance, where transparent and accountable decisions are necessary (Guidotti et al., 2018). While deep neural networks are powerful, Petch et al. (2021) point out that their black-box nature limits interpretability, hindering trust in sensitive applications. White-box models are more interpretable (Loyola-González, 2019) and, as Verenich et al. (2019) discusses, are often better suited for regulated settings; however, they frequently underperform compared to black-box models on complex, high-dimensional tasks (Aragona et al., 2021; Ding et al., 2019).

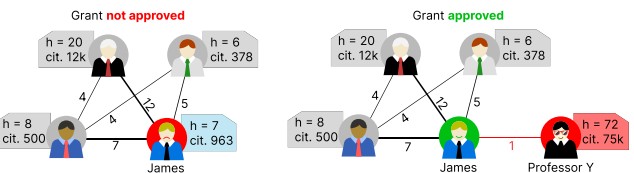

Figure 1: (left) James works with his research lab in publishing scientifically good papers with decent impact for the community; however, his grant application is not approved. (right) James writes a single, not-so-interesting paper with Professor Y, who happens to be highly influential within a broader community, which helps make James's grant application more likely to be approved. *James is happy, but at what cost?*

GNNs have gained significant attention (Wei et al., 2022), but like other deep learning models, they lack interpretability. To mitigate this, post-hoc methods, particularly counterfactual (CF) explanations, aim to reveal how input changes affect predictions. Recently, Graph Counterfactual Explainability (GCE) has recently emerged as a key area (Prado-Romero et al., 2023b). Consider a

research collaboration network, where nodes represent scientists, and edges indicate co-authorships on published papers. Suppose a young researcher, James, is recommended for a prestigious grant based on their position in the collaboration graph. A counterfactual explanation might state:

> *"If James had not co-authored a paper with Professor Y, he would not have been considered for the grant."*

This explanation highlights how specific connections within the graph structure influence high-stakes decisions. The counterfactual framework reveals how adding or removing certain collaborations alters the model's prediction, *offering insights into network-driven career advancements and systemic biases in academic recognition* – see Figure 1.

GCE techniques can be broadly categorized into search-based, heuristic-based, and learning-based approaches (Prado-Romero et al., 2023b). Search-based methods identify counterfactuals by searching within the existing data distribution. Heuristic-based approaches modify the input graph $G$ to create a perturbed version $G'$, ensuring that a prediction model $\Phi$ produces different classifications ($\Phi(G) \neq \Phi(G')$). These methods, however, require carefully designed heuristics, which often rely on domain expertise. For instance, generating chemically valid counterfactuals for molecular graphs necessitates an understanding of atomic valences and bonding rules. Learning-based strategies, in contrast, automate the discovery of meaningful perturbations by training on data, allowing for the generation of counterfactual examples at inference time.

In this work, we introduce XPlore, a substantial advancement over Lucic et al. (2022), designed to enhance counterfactual generation for GNNs. Unlike the original method, which modifies graphs solely through edge deletion, XPlore introduces additional flexibility by incorporating edge additions and node feature perturbations. This expansion allows for a more comprehensive exploration of graph modifications, improving the quality of counterfactual examples. Our approach employs a specialized loss function that autonomously guides perturbations, minimizing unnecessary modifications, a huge improvement over heuristic-driven methods. Moreover, rather than incorporating generative modeling into the learning objective, XPlore directly leverages gradient-based optimization to identify the closest counterfactual instance efficiently. This results in greater transparency: under access to oracle gradients, the entire explainer remains fully interpretable, avoiding the additional black-box component, and keeping the explanation pipeline lightweight, faithful to the classifier's decision boundary, and easier to audit.

Our contributions are as follows. (1) **Comprehensive Input Perturbation:** Our framework supports edge additions, edge removals, and node feature perturbations, enabling perturbation of nearly all input degrees of freedom. This richer perturbation space allows us to explore how structural and attribute-level changes jointly influence the oracle's prediction process and to uncover richer, truly counterfactual motifs that edge-only methods cannot express. (2) **Gradient-Guided Optimization:** By exploiting the properties of the loss, we find counterfactuals that are locally closest in the loss-optimization landscape, as guided by directed gradient-based modifications (see Appendix A). This guarantees that the counterfactual explanation is not only minimal under the objective but also consistent with the oracle's learned decision boundaries. (3) **Mitigation of Out-of-Distribution Issues:** We introduce a cosine similarity metric to quantify both structural and semantic fidelity of counterfactuals. Together with the use of edge additions and feature perturbations, this mitigates—but does not solve—out-of-distribution artifacts (Chen et al., 2023) and preserves class-relevant discriminative cues. Discussion follows in section 4.3. (4) **Domain coverage and Strong Empirical Performance:** We evaluate our method on a diverse suite of real-world and synthetic graph benchmarks and show that it consistently outperforms state-of-the-art methods in validity and fidelity while maintaining competitive runtime and semantic coherence.

## 2 RELATED WORK

We distinguish between inherently explainable and black-box methods, focusing on counterfactual explanations for graph classification. While counterfactuals are well-established in images and text (Vermeire et al., 2022; Zemni et al., 2023), their application to graphs is less common (Liu et al., 2021; Ma et al., 2022; Nguyen et al., 2022; Numeroso & Bacciu, 2021; Tan et al., 2022). As categorized in Prado-Romero et al. (2023b), graph counterfactual explanation methods fall into heuristic search and learning-based approaches. This work focuses on instance-level, learning-based explain-

ers that identify minimal perturbations to flip a prediction, providing actionable, data-point-specific insights.

Learning-based strategies often use perturbation matrices (Tan et al., 2022), reinforcement learning (RL) (Numeroso & Bacciu, 2021), or generative models (Ma et al., 2022; Prado-Romero et al., 2024). Some notable methods include CF-GNNExp. (Lucic et al., 2022), which learns a binary perturbation matrix; CF$^2$ (Tan et al., 2022), which balances factual and counterfactual reasoning through multi-objective optimization; and CLEAR (Ma et al., 2022), which uses a Variational Autoencoder (VAE) to generate counterfactuals. Other approaches, such as RSGG-CE (Prado-Romero et al., 2024) and D4Explainer (Chen et al., 2023), rely on advanced techniques like Generative Adversarial Networks (GANs) and discrete denoising diffusion, respectively. Additional methods are INDUCE (Verma et al., 2024) which is based on a RL-based inductive approach; COMBINEX (Giorgi et al., 2025) and C2Explainer (Ma et al., 2025). Emerging research on time-related graph counterfactuality (Prenkaj et al., 2024; Qu et al., 2024) is outside the scope of our work.

Differently from SoTA, we perform graph- and node-level explanations. In the former, we identify changes needed to alter the prediction for an entire graph. In the latter, we leverage loss information from individual nodes, allowing us to perturb both the local graph structure and node features with the targeted objective of changing a single node's label. Here, an oracle performs node-wise classification, and our method focuses on modifying the inputs that directly influence the prediction of the targeted node.

## 3 METHOD

### 3.1 PROBLEM FORMULATION

**Graph Counterfactual Explanation.** Suppose we have a well-trained GNN serving as an oracle $\Phi : \mathcal{G} \rightarrow \mathcal{Y}$ and an original graph instance $G \in \mathcal{G}$, with predicted label $\Phi(G)$. The objective of counterfactual explanation is to find a perturbed graph $G'$, obtained by a counterfactual model $\mathcal{E} : \mathcal{G} \rightarrow \mathcal{G}$, that minimally deviates from $G$ while ensuring that the oracle's prediction changes, i.e., $\Phi(G') \neq \Phi(G)$ (Prado-Romero et al., 2023b). Let $\Delta(G, G')$ denote the distance function measuring the difference between $G$ and $G'$; then the problem can be stated as:

$$G^* = \underset{G' \in \mathcal{G}'}{\arg\min} \, \Delta(G, G') \text{ s.t } \Phi(G) \neq \Phi(G'). \tag{1}$$

As done in Lucic et al. (2022), we generate counterfactuals by reformulating the above hard-constrained formulation into a soft-unconstrained optimization by minimizing the loss function:

$$L(G, G') = L_{\text{pred}}(G, G' \mid \Phi) + \beta \, L_{\text{dist}}(G, G'). \tag{2}$$

Here, $L_{pred}$ is a prediction loss that encourages $\Phi(G') \neq \Phi(G)$ and $L_{dist}$ is a distance loss that promotes similarity between $G'$ and $G$, with the trade-off controlled by $\beta$. Thus, the optimal counterfactual example for $G$ is obtained by solving Equation (2).

Table 1: Notation used in the Section 3.

| Symbol | Description |
|--------|-------------|
| $\mathcal{E}, \Phi$ | Counterfactual and oracle model |
| $G, G'$ | Original and counterfactual graphs |
| $A, \hat{A}$ | Adjacency matrices (original and counterfactual) |
| $P, \bar{P}$ | Original and new perturbation matrices for edges |
| $N, \bar{N}$ | Binary and continuous node feature perturb. matrices |
| $X, W$ | Node feature and weight matrices |
| $\bar{D}$ | Degree matrix |
| $\Gamma$ | Probability weight matrix with values $\gamma_{i,j} \in [0, 1]$ |
| $\sigma(\cdot)$ | Sigmoid activation function |
| softmax$(\cdot)$ | Softmax function |
| $\mathcal{T}_\alpha(\cdot)$ | Entry-wise threshold: $\mathcal{T}_\tau(X) = \mathbb{1}\{\sigma(X) \geq \alpha\}$ |
| $\odot$ | Element-wise (Hadamard) product |
| $\mathbb{1}\{\cdot\}$ | Indicator function |
| $K$ | Number of optimization iterations (e.g. 50, see C.2) |
| $L_{pred}, L_{dist}$ | Prediction and distance losses |

**Node Counterfactual Explanation.** Our method can be extended to node-level counterfactual explanations. In this scenario, rather than modifying only the target node, we perturb the entire graph, including all node features. Moreover, the loss formulation can be adapted to encompass a set of nodes by imposing a soft constraint that preserves the class labels of nodes other than the target through the use of binary cross-entropy with logits as the loss function. This framework naturally generalizes to multi-node classification tasks, where the objective is to generate counterfactual explanations for a set of nodes rather than a single node.

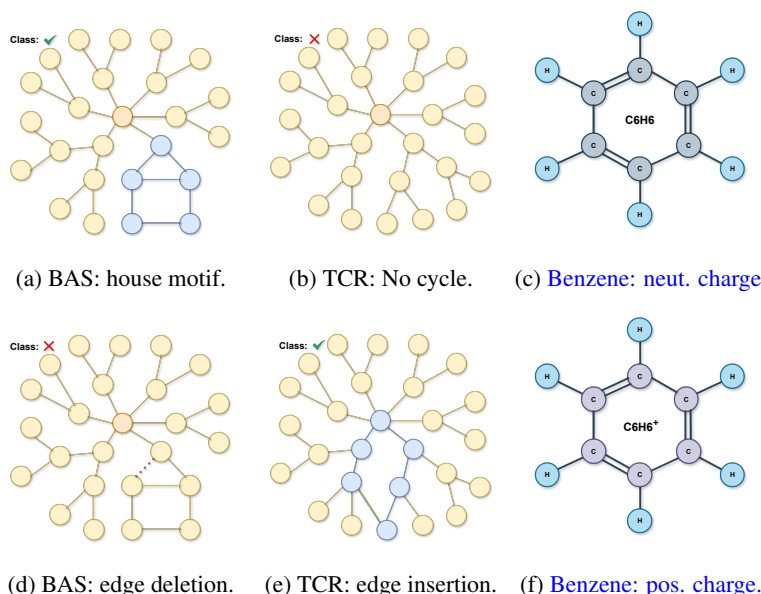

(a) BAS: house motif.    (b) TCR: No cycle.    (c) Benzene: neut. charge.

(d) BAS: edge deletion.    (e) TCR: edge insertion.    (f) Benzene: pos. charge.

Figure 2: Illustration of counterfactual explanations in graph classification. *a* and *d*: The class changes only after an **edge deletion** (red, dashed). *b* and *e*: the predicted class changes only after an **edge addition** (green). *c* and *f*: Feature perturbation of the carbon-group illustrates benzene converting from neutral to cationic by electron loss. XPlore enables these types of perturbations, alone or in combination, offering a broader search space than edge deletion only methods.

### 3.2 PROPOSED METHOD

We propose XPlore to sieve the less-constrained counterfactual search space compared to the base model (Lucic et al., 2022), aiming at better explanations while reducing the out-of-distribution effect shown in Chen et al. (2023). XPlore, not only drops instance edges but also adds them, enabling the discovery of counterfactual explanations for which a class change is unattainable by edge removal alone (e.g. TCR in Section B; see Figure 2). Moreover, by modifying node features, our approach can identify counterfactuals where a class change is unachievable solely through edge modifications (or any amount thereof), but rather via feature adjustments. We explored two modalities for handling features: i.e., **(1)** applying a gating mechanism to retain or discard node features, similarly to the edge-dropping process introduced in Lucic et al. (2022); **(2)** allowing for continuous adjustment in node features, enabling their values to increase/decrease smoothly. We provide formal guarantees on convergence, on $\ell_1$-minimality of perturbations and semantic fidelity in Appendix A.

**Original Adjacency matrix perturbations.** Let $A \in \{0,1\}^{n \times n}$ be the adjacency matrix – a square matrix whose elements indicate whether pairs of vertices are adjacent or not in the graph. We initially define the perturbation similarly to Lucic et al. (2022), having the CF generated matrix $\hat{A} = P \odot A$, with $P = \mathbf{1}_{n \times n}$, a binary perturbation matrix full of ones, with same dimensionality of $A$; and $\odot$ denoting element-wise (Hadamard) product (a real-valued $\bar{P}$ is first generated, then a threshold with a entry-wise sigmoid transformation $\sigma(\cdot)$ is performed to obtain a binary $P$, either discarding or retaining edges: $\mathcal{T}_{0.5}(\bar{P}) = \mathbb{1}\{\sigma(\bar{P}) \geq 0.5\}$). To include self message passing, we add self-loops in the form of the identity matrix during CF search. We define $X$ as the node feature matrix (so that $G = (A, X)$), $W$ as the weight matrix and $\bar{D}$ as the degree matrix, a diagonal matrix containing the number of edges attached to each vertex, based on $P \odot A + I$. The original goal is to only remove edges zeroing out entries in the adjacency matrix, so find $P$ (which acts as a gate) that minimally perturbs $A$ and use it to compute $\hat{A}$. Hence, the original counterfactual generating model, parameterized by $P$, is:

$$\mathcal{E}(A, X, W; P) = \mathrm{softmax}\left[\bar{D}^{-\frac{1}{2}}(P \odot A + I)\bar{D}^{-\frac{1}{2}}XW\right], \tag{3}$$

note that here, differently than Lucic et al. (2022), we perform the CF search not on a subgraph neighbourhood of a node, but rather on the whole graph. For undirected graphs, only the upper triangular part of $\bar{P}$ is parametrized and it is symmetrized at each step to obtain $\hat{A}$. This ensures that adjacency perturbations remain symmetric.

**Edge-masking with Free Insertions** The original formulation of the framework has some downsides when it comes to updating $P$: when the oracle prediction is obtained with the associated loss, and new edges need to be added. Essentially, gradients for adjacency and perturbation matrix entries are computed and flow back, but edges not present in $A$ ($A_{ij} = 0$) cause respective gradients of $P$ to be zeroed out ($\frac{\partial \mathcal{L}}{\partial P_{ij}} = \frac{\partial \mathcal{L}}{\partial \hat{A}_{ij}} \cdot \frac{\partial \hat{A}_{ij}}{\partial P_{ij}}$, $\hat{A}_{ij} = A_{ij} \cdot P_{ij} \Rightarrow \frac{\partial \hat{A}_{ij}}{\partial P_{ij}} = A_{ij}$.) *– As a results edges not initially present in A cannot be inserted during CF search by design choice, and gradients of missing edges indicating the direction of optimal change, even if are back-propagated until P, will not lead to any update to it, thus introducing confusion for other updates that may be co-adapted/correlated to them.* To enable the inclusion of any edge, we swap the roles of $A$ and $P$: initializing $\bar{P} \leftarrow A$, hence making it store the original adjacencies, but making it able to be updated by the iterative gradients; and making $\bar{A} = \mathbf{1}_{n \times n}$ be a matrix full of ones, fully connecting the graph. We account for the original missing edges by removing them from $\bar{P}$ (setting their values to zero), now $\hat{A}$ is computed as $\hat{A} = \bar{P} \odot \bar{A}$ plus self loops. By this swap, we give the explainer the possibility to freely add edges by updating the whole graph accordingly to the loss. $\bar{A}$ now becomes a (redundant) matrix of ones whereas $\bar{P}$ stores the graph connections, ready to be perturbed. We add to $P$ a small Gaussian noise ($std : \sigma = 0.1; \mu = 0$) to break the symmetry in the gradient updates.

We also assign a probability weight to missing edges of $\bar{P}$: we define a matrix $\Gamma \in [-1,1]^{n \times n}$. $\Gamma$ is added to $\bar{P}$ at initialization: $\bar{P} \leftarrow \bar{P} + \Gamma$. The idea is to manually adjust each $\gamma_{ij} \in [-1,1]$ so that by adding $\Gamma$ it is possible to induce edge presence or absence. For example we may be interested in a counterfactual explanation that has specific connections: we can directly encode them in $\Gamma$. Note that we swapped $P$ and $A$ to be coherent with the original formulation: as $A$ now consists of a matrix of ones, it is redundant for updating $\bar{P}$ and can

Table 2: XPlore steps to perturb edges and node features.

| Step | Action |
|------|--------|
| 1 | Initialize $\bar{P} \leftarrow A + \mathcal{N}(0, 0.1^2)$ and optionally add $\Gamma$. |
| 2 | Initialize $\bar{N} \leftarrow \mathbf{1}_{n \times f}$ |
| 3 | Compute $\hat{A} = max(\mathcal{T}_{0.5}(\bar{P}), I)$ |
| 4 | Compute contribution of $\hat{A}$ and $\bar{N}$ to $L_{pred}$ through the oracle's prediction (Equations (6) and (7)). |
| 5 | Update $\bar{P}$ and $\bar{N}$ via gradient descent. |

be simply omitted. Now, the explainer is as in Equation (4):

$$\mathcal{E}(X, W; \bar{P}) = \mathcal{S}\left[\bar{D}^{-1/2} \hat{A} \bar{D}^{-1/2} XW\right], \tag{4}$$

where $\mathcal{S}$ is either the element-wise sigmoid or the softmax function, depending on whether the task is binary or multi-class; and $\bar{D}$ is based on $\hat{A}$, slightly different from Equation (3), ensuring that values in the diagonal are ones, specifically when self-loop are already present – due to $\hat{A}$ having no more the diagonal entries fixed at zero as it depends on $\bar{P}$ for storing perturbed adjacencies and not on $A$ any more.

**Node Features Perturbation.** Similarly to what has been done for the edge perturbation matrix $\bar{P}$, we can introduce a perturbation matrix $\bar{N} = \mathbf{1}_{n \times f}$, $f$ number of node features, to perform continuous perturbation on the node features. We can apply two different mechanisms (Algorithm 1): i.e., **(1)** gate the features values with a sigmoid, by applying the sigmoid and then the threshold to have a binary mask, and **(2)** let them freely change, updating them proportionally to the gradient information of the loss. In results, these mechanisms added on top of the edges addition, are defined as XPlore w/ gating and XPlore w/ freedom. Hence, the final formulation of our explainer considering all perturbations is:

$$N = \mathcal{T}_{0.5}(\bar{N}) \text{ if } gate \text{ else } \bar{N} \tag{5}$$

$$\mathcal{E}(X, W; \bar{P}, \bar{N}) = \mathcal{S}\left[\bar{D}^{-1/2} \hat{A} \bar{D}^{-1/2} (X \odot N)W\right]. \tag{6}$$

**Loss function optimization.** We generate $\bar{P}$ by minimizing Equation (2), defining the prediction loss $L_{\text{pred}}$ as in Equation (7): for $L_{logits}$ we employ the cross-entropy (CE) loss for single-label

classification tasks, encompassing binary and multi-class scenarios, and the binary cross-entropy with logits loss for multi-label classification tasks:

$$L_{\text{pred}}(G, G' \mid \Phi) = -\mathbf{1}\big[\Phi(G) = \Phi(G')\big]$$
$$\cdot L_{\text{logits}}(\Phi_{\ell-1}(G), \Phi_{\ell-1}(G')), \tag{7}$$

where $\Phi_{\ell-1}(G)$ outputs the logits given the input $G$, and $\ell$ is the number of layers. $G'$ is $\mathcal{E}(G)$. The $L_{dist}$ in Equation (2) is the element wise distance between $G$ and $G'$, corresponding to the sums of the two $L_p$-norms of $A$ and $A'$, and $X$ and $X'$:

$$L_{\text{dist}}(G, G') = \|A - A'\| + \|X - X'\|. \tag{8}$$

## 3.3 Algorithmic Implementation

---

**Algorithm 1** XPlore.

---

1: **Input** graph $G = (A, X)$, trained oracle $\Phi$, explainer $\mathcal{E}$, loss function $L$, learning rate $\alpha$, number of iterations $K$, matrix $\Gamma$, node features $gate$ flag, distance function $d$.
2: $y \leftarrow \Phi(G)$
3: $\bar{P} \leftarrow A + \mathcal{N}(0, 0.01) + \Gamma$
4: $\bar{N} \leftarrow \mathbf{1}_{n \times f}$
5: $G^* = [\,]$
6: **for** $K$ iterations **do**
7: $\quad G', G^* = $ GET_CF_EXAMPLE()
8: $\quad L \leftarrow L(G, G', \Phi)$ {▶ Eqns. 7-8}
9: $\quad \bar{P} \leftarrow \bar{P} - \alpha \nabla_{\bar{P}} L$ {▶ Update $\bar{P}$}
10: $\quad \bar{N} \leftarrow \bar{N} - \alpha \nabla_{\bar{N}} L$ {▶ Update $\bar{N}$}
11: **return** $G^*$
12:
13: **func** GET_CF_EXAMPLE()
14: $\quad \hat{A} \leftarrow max(\mathcal{T}_{0.5}(\bar{P}), I)$ {▶ Perturbed adj.}
15: $\quad$ **if** $gate$ **then** {▶ Gate node feat.}
16: $\quad\quad N \leftarrow \mathcal{T}_{0.5}(\sigma(\bar{N}))$
17: $\quad$ **else** {▶ Freely perturb node feat.}
18: $\quad\quad N \leftarrow \bar{N}$
19: $\quad N \leftarrow N \odot X$
20: $\quad G'_{cand} \leftarrow (\hat{A}, N)$
21: $\quad$ **if** $\Phi(G) \neq \Phi(G'_{cand})$ **then**
22: $\quad\quad G' \leftarrow G'_{cand}$
23: $\quad\quad$ **if** not $G^*$ **then**
24: $\quad\quad\quad G^* \leftarrow G'$ {▶ First CF}
25: $\quad\quad$ **else if** $d(G, G') \leq d(G, G^*)$ **then**
26: $\quad\quad\quad G^* \leftarrow G'$ {▶ Closer CF}
27: $\quad$ **return** $G', G^*$

---

We summarize the details of our method XPlore[1] in Algorithm 1. Given a graph in the test set G, its prediction is obtained from the GNN oracle $\Phi$. $\bar{P}$ is initialized as the adjacency matrix $A$ and summed with $\Gamma$, giving missing edges a probability value of $\gamma_{i,j} \in [-1, 1]$. $\bar{N}$ is assigned to a matrix of ones. XPlore is run for $K$ iterations (see AppendixC.2), at each one, an optimization step is performed to find a valid counterfactual and improve it, there is no stopping criterion as the same CF is iteratively modified and potentially improved across successive iterations, as conceived by Lucic et al. (2022). Equation (6) is used to find a counterfactual example. $P$ is computed by applying a sigmoid transformation on $\bar{P}$ and then a threshold to obtain a binary matrix. Although hard threshold $\mathcal{T}$ is applied in forward pass, gradients are back-propagated through this non-differentiable step via the straight-through estimator (STE) (Bengio et al., 2013), which treats thresholding as identity during backpropagation (cf. Appendix A.2). We add self-loops by summing the identity matrix. According to the node perturbation mechanism, node features are either gated or simply multiplied by the node features perturbation matrix $\bar{N}$. The candidate graph produced $G'_{cand}$ is fed to the GNN oracle $\Phi$. If the output prediction is different from the initial node, a valid counterfactual is found. The closer counterfactual is returned as the optimal CF example $G^*$ after $K$ iterations upon success. $\bar{P}$ and $\bar{N}$ are updated based on the loss, which is calculated according to Equations (2) and (6) to (8). This setting allows to perform updates freely by perturbing edges and node features and adding missing ones. The optimal explanation is retrieved as $\Delta_G^* = G - G^*$. The algorithm is linear in the number of edges and in the multiplication of features with number of nodes (i.e. $O\big(|E|\, d + n\, d\, f\big)$, see Appendix A.7).

# 4 Experiments

## 4.1 Experimental Setup

We train separate GCN oracles for each dataset using an 80-20 train-test split, with architectural and hyperparameter details provided in Appendices C.1 and C.2, respectively. We compare XPlore

---

[1] https://anonymous.4open.science/r/XPlore

Table 3: Comparison of XPlore with SoTA methods (validity ↑ – up; fidelity ↑ – down). **Bold** is best-performing; underline is second-best. **XPlore is best in 17/18 on validity, and 17/18 on fidelity (of which 1/17 on par with RSGG-CE).** Table 9 in the Appendix shows a detailed comparison on other metrics. † indicates the second-best explainer.

| | TCR | BAS | BZR | AIDS | ENZYMES | Fingerprint | COLORS-3 | TG | MUTAG | COX2 | BBBP | PROTEINS | COLLAB | TRIANGLES | DBLP | IMDB | TWITTER | MSRC |
|---|---|---|---|---|---|---|---|---|---|---|---|---|---|---|---|---|---|---|
| iRand | 27.92 | 50.70 | 27.16 | 0.00 | 26.67 | 0.09 | 42.99 | 36.16 | 2.66 | 69.81 | 19.76 | 18.87 | 4.60 | 6.38 | 1.17 | 3.70 | 0.00 | 95.74 |
| CF² | 50.04 | 45.78 | 19.75 | 0.10 | 68.33 | 24.52 | 52.07 | 49.86 | 0.00 | 24.20 | 25.26 | 16.35 | 52.66 | 37.13 | 5.76 | 50.60 | 38.82 | 90.94 |
| CLEAR | 50.68 | 50.96 | 60.49 | 16.75 | 83.17 | 72.73 | 0.00 | 58.40 | 35.11 | 22.06 | 22.90 | 0.00 | 0.00 | 89.99 | 0.68 | 56.20 | 7.65 | 23.98 |
| RSGG-CE † | 67.90 | 91.04 | 21.23 | 19.80 | 98.33 | 90.46 | 94.57 | 89.28 | 56.91 | **99.36** | 22.90 | 58.67 | 0.00 | 99.84 | 57.51 | **86.10** | 48.36 | **100.0** |
| D4Explainer | 44.82 | 35.16 | 20.00 | 0.10 | 68.00 | 24.52 | 0.00 | 49.86 | 9.57 | 22.06 | 21.24 | 0.00 | 0.00 | 31.11 | 7.02 | 49.60 | 23.45 | 90.94 |
| CF-GNNExpl | 50.04 | 44.18 | 19.75 | 0.10 | 68.33 | 24.52 | 52.07 | 49.86 | 10.11 | 22.06 | 22.31 | 16.35 | 52.66 | 37.13 | 5.76 | 50.60 | 38.82 | 90.94 |
| XPlore | **100.0** | **100.0** | **100.0** | **32.30** | **100.0** | **100.0** | **100.0** | **100.0** | **67.55** | **99.36** | **81.51** | **65.41** | **100.0** | **100.0** | **91.84** | 78.80 | **50.71** | **100.0** |
| iRand | 0.279 | 0.507 | 0.262 | 0.000 | 0.238 | 0.001 | 0.300 | 0.356 | 0.005 | 0.677 | 0.275 | 0.183 | 0.006 | 0.053 | 0.06 | 0.026 | 0.000 | 0.391 |
| CF² | 0.500 | 0.457 | 0.188 | 0.001 | 0.633 | 0.133 | 0.398 | 0.499 | 0.005 | 0.229 | 0.253 | 0.151 | 0.262 | 0.364 | 0.027 | 0.366 | 0.345 | 0.371 |
| CLEAR | 0.507 | 0.510 | 0.595 | 0.168 | 0.797 | 0.515 | 0.000 | 0.584 | 0.309 | 0.208 | 0.229 | 0.000 | 0.000 | 0.892 | 0.005 | 0.420 | 0.039 | 0.012 |
| RSGG-CE † | 0.679 | 0.910 | 0.202 | 0.198 | 0.930 | 0.653 | 0.824 | 0.888 | 0.516 | **0.968** | 0.229 | 0.556 | 0.000 | 0.987 | 0.450 | **0.664** | 0.434 | **0.423** |
| D4Explainer | 0.448 | 0.446 | 0.190 | 0.001 | 0.632 | 0.133 | 0.000 | 0.499 | 0.021 | 0.208 | 0.212 | 0.000 | 0.000 | 0.302 | 0.031 | 0.358 | 0.161 | 0.371 |
| CF-GNNExpl | 0.500 | 0.442 | 0.188 | 0.001 | 0.633 | 0.133 | 0.398 | 0.499 | 0.005 | 0.208 | 0.223 | 0.151 | 0.262 | 0.364 | 0.027 | 0.366 | 0.345 | 0.371 |
| XPlore | **1.000** | **1.000** | **0.980** | **0.323** | **0.942** | **0.730** | **0.896** | **0.994** | **0.548** | **0.968** | **0.803** | **0.627** | **0.570** | **0.988** | **0.748** | 0.606 | **0.613** | **0.423** |

against CF-GNNExpl. (Lucic et al., 2022), CF² (Tan et al., 2022), CLEAR (Ma et al., 2022), RSGG-CE (Prado-Romero et al., 2024), D4Explainer (Chen et al., 2023), and iRand (Prado-Romero et al., 2023a). For iRand, we set the edge perturbation probability ($p$) to 0.01 and the number of iterations ($t$) to 3.

Following the protocol from Prado-Romero et al. (2023b), we use a comprehensive evaluation suite, including *Oracle Accuracy*, *Validity*, *Fidelity*, *Sparsity*, *Graph Edit Distance (GED)*, *Oracle Calls* and *Runtime*. Additionally, we use the cosine similarity (CS), which measures the semantic similarity between original and counterfactual graphs. Unlike structural metrics such as GED or sparsity, CS captures alignment in graph meaning by comparing embeddings.

Given a set of graph embedders $E$ ($|E| = M$), we compute vector embeddings $\mathbf{e_{ji}} = E_j(G_i)$ and $\mathbf{e'_{ji}} = E_j(G'_i)$ for each graph $G_i$ ($|G| = N$) and its counterfactual $G'_i$, using $E_j \in E$. CS is defined as:

$$\text{CS}(\mathbf{e_{ji}}, \mathbf{e'_{ji}}) = \frac{1}{MN} \sum_{j=1}^{M} \sum_{i=1}^{N} \frac{\mathbf{e_{ji}} \cdot \mathbf{e'_{ji}}}{|\mathbf{e_{ji}}||\mathbf{e'_{ji}}|} \in [-1, 1] \quad (9)$$

The set $E$ includes following embedders: Feather-G (Rozemberczki & Sarkar, 2020), Graph2Vec (Narayanan et al., 2017), NetLSD (Tsitsulin et al., 2018), WaveletCharacteristic (Wang et al., 2021), IGE (Galland & Lelarge, 2019), LDP (Cai & Wang, 2022), GeoScattering (Gao et al., 2019), GL2Vec (Chen & Koga, 2019), SF (de Lara & Pineau, 2018), and FGDS (Verma & Zhang, 2017). Definitions for standard metrics are detailed in Appendix C.3.

## 4.2 EXPERIMENTAL RESULTS

**XPlore on average achieves +15.1% percentage improvement in validity, and +14.0% on fidelity on across the board vs. the second-best in graph classification.** Table 3 reports the performance of XPlore and all state-of-the-art baselines across ten runs on each dataset. See table 9 in Appendix for a broader analysis. We show the validity, fidelity, sparsity, and oracle calls since they describe the goodness of the explainer in finding valid counterfactuals that are also cheap in terms of querying the underlying predictor. Note that XPlore is the best in 17/18 datasets in terms of validity and 17/18 regarding fidelity. This shows that a simple modification to the learning objective and node feature manipulation enhances the faithfulness of the explainer. XPlore maintains a relatively low number of oracle calls, indicating that in at most 14.692 (see TG) iterations, the optimal counterfactual is found (see line 6 of Algorithm 1). Additionally, we show the relationship between GED and CS in Figure 3b.

However, we are interested in those explainers that might induce more edit changes to the counterfactual yet maintaining high semantic similarity. Note that XPlore generally exhibits high GED values although it consistently achieves strong CS scores, underlining our intuition that semanticity plays an important role

Table 4: Validity and Fidelity results for node explanation.

| Method | BAS | | BZR | | ENZYMES | | MSRC | | TWITTER | |
|---|---|---|---|---|---|---|---|---|---|---|
| | Val. ↑ | Fid. ↑ | Val. ↑ | Fid. ↑ | Val. ↑ | Fid. ↑ | Val. ↑ | Fid. ↑ | Val. ↑ | Fid. ↑ |
| CF-GNNExpl | 2.42 | 0.024 | 7.21 | 0.071 | 12.24 | 0.001 | 7.23 | 0.000 | 26.50 | 0.000 |
| XPlore | **4.74** | **0.047** | **99.91** | **0.998** | **81.06** | **0.695** | **81.40** | **0.753** | **100.0** | **0.034** |

in counterfactuality. We reserve more thorough experimentation for future work. Lastly, we show the runtime (s) at inference time for all methods to show that XPlore is at least on par with SoTA explainers (see Figure 4 in the Appendix).

**XPlore outperforms CF-GNNExpl on node classification, improving per-class explainability by +62.30 validity points on average.** We compare both methods on five datasets, selecting one per domain. Despite prior claims (Lucic et al., 2022), generating effective node-level counterfactuals remains difficult, particularly in class-specific scenarios. As summarized in Table 4, XPlore shows consistent improvements, reflecting its ability to finely manipulate node-level features.

### 4.3 RESIDUAL OUT-OF-DISTRIBUTION INFLUENCE

Chen et al. (2023) observed that CF-GNNExpl leverages the out-of-distribution (OOD) effect that influences the oracle's prediction, by deriving explanatory subgraphs while omitting additional potential edges.

Consequently, the extracted explanation lacks discriminative information for the CF class but is still classified as CF due to this OOD effect, ultimately misleading the oracle and compromising reliability. As shown in Figure 3b, we reduce OOD effect exploitation by increasing cosine similarity: XPlore semantic similarity is higher w.r.t. competitors with similar GED scores and operating at the same perturbation level , thus relying less on OOD effect. Figure 5 in Appendix shows that by identifying more CFs, XPlore also captures harder instances, yielding elongated or bimodal CS distributions, reflecting competitive performance on both easy and hard instances. t-SNE projection of the Wavelet Characteristic embedding space reveals that XPlore's CF exhibits substantial semantic overlap with the target class, landing correctly in the Cycle distribution embedding space, which integrates both topological and node attribute information (Figure 3a). Other explainers do not achieve this behavior, remaining within the Tree distribution and exploiting OOD effect. Although XPlore's ability leads to generate more comprehensive and hence robust explanations, the OOD effect remains present (see Appendix C.4), this highlights the oracle's limited expressive power and the fact that it was not trained to mitigate adversarial instances.

We refer the reader to Leemann et al. (2024) for non-adversarial CF explanations.

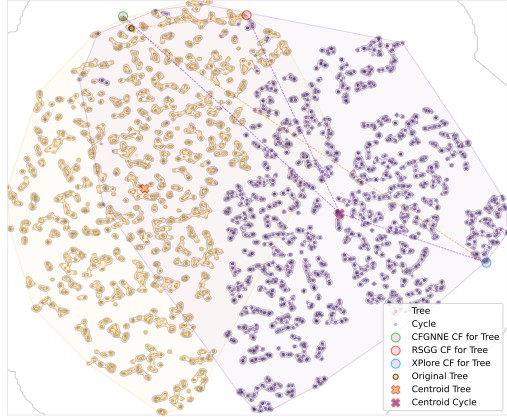

(a) t-SNE projection of Wavelet Characteristic embeddings for TCR, comparing CFs generated by CF-GNNExpl, XPlore and RSGG for the Tree motif. CF-GNNExpl and RSGG find a close CF but fail to land in the Cycle distribution, while XPlore achieves this correctly.

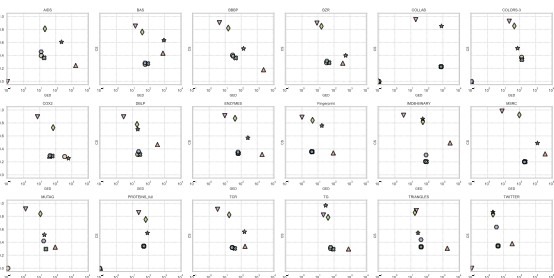

(b) Relationship between GED and CS. **XPlore consistently achieves strong CS scores w.r.t. the competitors along-side same GED values in log-scale**, indicating that counterfactuals retain meaningful semantic relations to original graphs, partially mitigating OOD.

Figure 3: OOD Effect and GED-CS Relationship.

## 4.4 Ablation Studies

**Free node perturbation leads to better counterfactual validity.** Since XPlore performs free node perturbations, we study the effect of this freedom by constraining it and assessing whether simpler mechanisms work in the same manner. Therefore, we test **(1)** XPlore w/ gating that performs edge additions and node feature manipulation by gating them, similarly to the original edge drop mechanism (Lucic et al., 2022), either retaining or discarding them; and **(2)** a variant that does not have the gating mechanism and cannot freely manipulate node features, called XPlore w/o freedom & gating. To be consistent with the previous nomenclature, we rename XPlore in XPlore w/ freedom. We show the validity of these variants in Table 5. Thanks to the better manipulation of graph elements, all three variants have higher validity w.r.t. the original CF-GNNExplainer. The additional ability to alter node features generally leads to even better results. The flexibility given to this modulation allows to tune XPlore to match the explainability criteria and find better counterfactuals.

Table 5: Validity (%) for different variants of XPlore ($\gamma = 0.01$). Bold values are best.

|  | w/o freedom + gating | w/ gating | w/ freedom | CF-GNNExpl |
|---|---|---|---|---|
| TCR | 99.680 | 65.580 | **100.00** | 50.040 |
| TG | 99.660 | **100.00** | **100.00** | 49.860 |
| BAS | 94.340 | **100.00** | **100.00** | 44.180 |
| MUTAG | 45.745 | 60.106 | **67.553** | 10.106 |
| BZR | 98.519 | **100.00** | **100.00** | 19.753 |
| COX2 | **99.358** | 97.859 | 98.073 | 22.056 |
| AIDS | 0.300 | 19.850 | **30.300** | 0.100 |
| BBBP | 38.892 | 38.794 | **79.794** | 22.315 |
| ENZYMES | 76.500 | **99.667** | 98.833 | 68.333 |
| PROTEINS | 17.071 | 64.960 | **65.409** | 16.352 |
| Fingerprint | 26.105 | **98.232** | 94.742 | 24.523 |
| COLLAB | 55.640 | **100.00** | 93.380 | 52.660 |
| COLORS-3 | 67.943 | **100.00** | 94.476 | 52.067 |
| TRIANGLES | 39.873 | 90.436 | **100.00** | 37.127 |

In Table 6, we report metrics for $\gamma = 0$ and $\gamma = 0.01$ respectively, where $\gamma$ is the value used to populate the $\Gamma$ matrix entries that correspond to missing edges of $A$. $\Gamma$ is added to $P$ at initialization. Recall that an edge is present if and only if $\sigma(v_i, v_j) > 0.5$ s.t. $v_i, v_j$ are nodes. A positive $\gamma$ increases the likelihood that edges are present at initialization, since $\sigma(\gamma + \epsilon) > 0.5$. However, because of the added Gaussian noise, presence is not guaranteed. This behavior possibly results in a higher number of counterfactuals found, but at the cost of higher GED, higher sparsity, and lower CS. This functionality may be needed if some edge is required to be present in the CF explanation; this prior knowledge is injected in the form of edge probabilities in matrix $\Gamma$.

Table 6: Metrics comparison over TCR dataset for different values of hyperparameter $\gamma$. We emphasize in bold the variants of XPlore per metric; underlined is the second-best.

|  | Validity ↑ | | | Fidelity ↑ | | | Sparsity ↓ | | | GED ↓ | | | Oracle Calls ↓ | | | CS ↑ | | |
|---|---|---|---|---|---|---|---|---|---|---|---|---|---|---|---|---|---|---|
|  | $\gamma = 0$ | $\gamma = 0.01$ | $\gamma = 0.1$ | $\gamma = 0$ | $\gamma = 0.01$ | $\gamma = 0.1$ | $\gamma = 0$ | $\gamma = 0.01$ | $\gamma = 0.1$ | $\gamma = 0$ | $\gamma = 0.01$ | $\gamma = 0.1$ | $\gamma = 0$ | $\gamma = 0.01$ | $\gamma = 0.1$ | $\gamma = 0$ | $\gamma = 0.01$ | $\gamma = 0.1$ |
| w/o freedom + gating | 50.04 | 99.68 | 70.86 | 0.500 | 0.997 | 0.709 | 0.745 | 3.578 | 3.996 | 41.00 | 201.6 | 224.3 | 46.00 | 24.09 | 16.16 | 0.309 | 0.273 | 0.406 |
| w/ gating | 65.58 | 65.58 | 49.18 | 0.656 | 0.656 | 0.492 | **0.249** | 4.931 | 5.491 | **14.00** | 278.5 | 308.5 | 12.36 | 3.706 | **2.000** | 0.955 | 0.426 | 0.293 |
| w/ freedom | 96.00 | **100.0** | **100.0** | 0.960 | **1.000** | **1.000** | 0.251 | 2.708 | 4.743 | **14.00** | 151.02 | 264.1 | 5.881 | 3.593 | 3.570 | **0.956** | 0.562 | 0.337 |

## 5 Conclusion

We introduced XPlore, a novel counterfactual explainer for Graph Neural Networks (GNNs) that explores a more complete search space, allowing for both edge deletions/insertions and node feature perturbations. Our method is not a black box itself; it relies on a basic gradient-based optimization building block to perform an intuitive counterfactual search. Once an oracle is trained, our explainer is fast and lightweight, requiring no further training or intense computation. XPlore identifies minimal yet impactful modifications, ensuring high-quality counterfactual explanations while avoiding heuristic-driven biases. Through extensive benchmarking across various datasets, XPlore consistently outperformed existing state-of-the-art explainers in validity and fidelity. Our empirical evaluation highlights that XPlore achieves an average percentage gain of +17.3% in validity and +15.0% in fidelity over the second-best method across multiple datasets. We also rely on the cosine similarity of graph embeddings as a complementary metric, which demonstrates that XPlore captures structural modifications while maintaining semantic fidelity. This confirms the effectiveness of incorporating a more flexible perturbation space and a comprehensive loss function.

**Future work.** We acknowledge that mitigating out-of-distribution (OOD) effects and optimizing node feature perturbations remain key challenges. While our method has shown promise in this area, the capacity to reduce these effects is contingent on how effectively the oracle captures the data distribution. We posit that less robust oracles are more prone to OOD explanations, which gives us hope that a more robust oracle, such as one based on a diffusion model, could significantly reduce

XPlore's OOD effects. We will also investigate the link between counterfactual explanations and model robustness, aiming to create explainability methods that are both interpretable and resistant to adversarial manipulation. Another avenue would be that of refining the loss function to better balance minimal perturbations with explanation quality and developing more precise control over feature modifications.

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

## A  THEORETICAL FOUNDATIONS OF GRADIENT-GUIDED COUNTERFACTUAL PERTURBATIONS

This section is meant to give a mathematical justification for why our gradient-based procedure converges and finds small ($\ell_1$-minimal) local perturbations to the graph and features.

**Appendix Roadmap.** We first establish convergence of projected gradient descent (A.1), then handle non-smooth thresholding (A.2), prove the $\ell_1$-minimality bound (A.3), derive the edge-insertion/deletion condition (A.4), show our prediction loss is $L$-smooth (A.5), choose sparsity–accuracy trade-offs (A.6), analyze computational complexity (A.7), extend to weighted and directed graphs (A.8), and finally discuss search-space expressivity (A.9).

In Section 3, we described a soft, differentiable objective $L(\cdot)$ ( Equation (2)) that trades off between (a) changing the model's prediction and (b) paying an $\ell_p$ "cost" for every edit. What remains is to show that if we optimize this objective with a natural algorithm, we end up at a valid counterfactual that also doesn't make too-large edits.

Hence, we here justify why the joint gradient-based optimization over edge masks $P$ and feature perturbations $N$ (cf. Algorithm 1 and eqs. (2) and (6)) converges to a meaningful counterfactual and yields nearly $\ell_1$-minimal changes.

### A.1  CONVERGENCE TO LOCAL MINIMIZERS

Let $\theta = (\text{vec}(P), \text{vec}(N))$ and define the soft objective $L(\theta)$ as in Equation (2), with projected gradient descent (PGD) updates

$$\theta^{t+1} = \Pi_\Theta\big(\theta^t - \eta\nabla L(\theta^t)\big),$$

where the projection $\Pi_\Theta$ enforces $P \in [0,1]^{n\times n}$ and leaves $N \in \mathbb{R}^{n\times f}$ unconstrained.

**Assumption 1** *The loss $L$ is differentiable except for a convex nonsmooth part $L_{\text{dist}}$, $L$-smooth on the smooth part, and coercive on $\Theta$.*

Under these standard conditions and for step-size $\eta \leq 1/L$, classical PGD convergence results (Bertsekas, 1997, Ch. 2.3) guarantee that the iterates satisfy $\theta^{t+1} - \theta^t \to 0$, and any limit point $\theta^*$ is Clarke-stationary:

$$0 \in \nabla L(\theta^*) + NC_\Theta(\theta^*),$$

i.e., no feasible small perturbation can locally decrease $L$. Hence, PGD converges to a stationary point of the soft objective under the box constraint, as required for our method.

## A.2 HANDLING NON-SMOOTH THRESHOLDING VIA SURROGATE GRADIENTS

Let $P \in [0, 1]^{n \times n}$ be our continuous edge-importance matrix. We obtain a hard adjacency

$$\hat{A} = 1[P > 0.5] \in \{0, 1\}^{n \times n}$$

for the forward GNN pass. To propagate gradients back to $P$, we use the common PyTorch "detach" trick (an instance of the straight-through estimator (Bengio et al., 2013))

```
mask   = (P >= 0.5).float() # no gradient
P_hard = P+(mask - P).detach() # forward uses mask; backward flows into P
outputs = GNN(P_hard, features)
```

Listing 1: STE-based hard-thresholding in PyTorch

In effect, the backward pass treats the binarization as if it were the identity function. This can be seen as a limiting case of using a steep sigmoid surrogate $\sigma_\alpha(x) = 1/(1+e^{-\alpha(x-0.5)})$ with $\alpha \to \infty$. Under this STE, the composite loss

$$L(P) = L_{\text{pred}}(\hat{A}, X) + \beta||P - P_0||_1$$

is differentiable almost everywhere in $P$, and convergence to a Clarke-stationary point follows from standard projected-gradient arguments. $P_0$ is the original adjacency mask.

Note that STE is a heuristic used in practice, and the previous convergence proofs refer to the continuous/relaxed problem, not the exact gradients through hard thresholding.

Note that we replace the non-differentiable hard threshold by the identity in the backward pass (i.e. STE). Using the identity in the backward pass induces a bias (dependent on the size of the downstream gradient) by propagating nonzero gradients where the true hard-threshold has zero derivative, whereas zeroing those gradients would irreversibly freeze masked entries – hence, to mitigate this, one may use a temperature-controlled sigmoid $\sigma_\alpha$ (or even an annealed $\alpha$) that smoothly trades off bias and trainability.

## A.3 $\ell_1$-MINIMALITY BOUND OF PERTURBATIONS

Let again,

$$\theta = \big(\text{vec}(P), \text{vec}(N)\big), \quad \theta_0 = \big(\text{vec}(A), \text{vec}(1_{n \times f})\big),$$

and the well-known soft objective

$$L(G; \theta) = L_{\text{pred}}\big(\Phi(G), E(G; \theta)\big) + \beta||\theta - \theta_0||_1,$$

**Lemma 1** *Assume $L_{\text{pred}}$ is bounded below and set $\Delta = L_{\text{pred}} - \inf_\Theta L_{\text{pred}}$, the difference of $L_{\text{pred}}$ with the infimum (greatest lower bound) of the prediction loss $L_{\text{pred}}$ over all feasible $\theta \in \Theta = [0, 1]^{n \times n} \times \mathbb{R}^{n \times f}$, the best possible point in $\Theta$ achievable by $\theta^*$. Then any stationary point $\theta^*$ of $L$ satisfies*

$$||\theta^* - \theta_0||_1 \leq \frac{\Delta}{\beta},$$

*where $\theta_0$ is the original unperturbed $(P, N)$-vector.*

**Proof 1** *Since $\theta_0 \in \Theta$, stationarity (or just minimality) of $\theta^*$ gives*

$$L(\theta^*) \leq L(\theta_0) \implies L_{\text{pred}}(\theta^*) + \beta||\theta^* - \theta_0||_1 \leq L_{\text{pred}}(\theta_0).$$

*Rearranging,*

$$\beta||\theta^* - \theta_0||_1 \leq L_{\text{pred}}(\theta_0) - L_{\text{pred}}(\theta^*)$$
$$\leq L_{\text{pred}}(\theta_0) - \inf_\Theta L_{\text{pred}} = \Delta$$

*Dividing by $\beta$, we get the clean bound*

$$||\theta^* - \theta_0||_1 \leq \frac{\Delta}{\beta}.$$

*Thus, choosing $\beta \geq \Delta/\epsilon$ forces $||\theta^* - \theta_0||_1 \leq \epsilon$, (i.e. the size of our edits (in $\ell_1$) is at most the ratio of "how much we can lower the prediction loss" over "how costly each unit of edit is."*

**Practical choice of $\beta$.** In theory, the bound

$$\|\theta^* - \theta_0\|_1 \;\leq\; \frac{\Delta}{\beta}, \quad \Delta = L(\theta_0) - L(\theta^*)$$

guides $\beta$ to achieve a target $\ell_1$-norm (sparsity). In practice $\Delta$ is unknown, so we select $\beta$ empirically – e.g. via grid search or cross-validation – by monitoring the resulting $\|\theta^*\|_1$ (or edge count) and choosing the smallest $\beta$ that attains the desired sparsity level $\epsilon$.

A.4 EDGE-INSERTION AND EDGE-DELETION CONDITION

We now derive a simple, quantitative criterion under which a zero-entry $p_{ij} = 0$ will be driven strictly positive, and a one-entry $p_{ij} = 1$ will be driven to zero, by projected gradient descent on the soft objective **??**.
Recall that $L_{\text{dist}}$ penalizes deviations of both edges and node features, but here we focus on its effect on the edge variable $p_{ij}$.

We consider the soft objective

$$L(P, N) \;=\; L_{\text{pred}}(P, N) \;+\; \beta\, L_{\text{dist}}(P, N),$$

where

- $P = [p_{ij}] \in \{0, 1\}^{n \times n}$ is the *final* binary adjacency mask,
- $N$ is the node-feature matrix,
- $\bar{P} = [\bar{p}_{ij}] \in [0, 1]^{n \times n}$ is the *original* continuous adjacency (here we assume no noise, in practice we add a small Gaussian noise to $\bar{P}$, which does not materially change the $\pm 1$ subgradients at the boundaries),
- $L_{\text{dist}}(P, N)$ includes the edge penalty $\sum_{i,j} |p_{ij} - \bar{p}_{ij}|$, Equation (8) (plus any feature-penalty terms).

To decide whether an edge coordinate $p_{ij}$ will switch its value under projected gradient descent, we inspect the one-dimensional update

$$p_{ij}^{(t+1)} = \Pi_{\{0,1\}}\left(p_{ij}^{(t)} - \eta\left(g_{ij} + \beta\, d_{ij}\right)\right),$$

where

$$g_{ij} = \frac{\partial L_{\text{pred}}}{\partial p_{ij}}, \quad d_{ij} = \frac{\partial}{\partial p_{ij}}\left|p_{ij} - \bar{p}_{ij}\right| = \text{sign}\left(p_{ij} - \bar{p}_{ij}\right).$$

By the sign-definition,

$$d_{ij} = \begin{cases} +1, & p_{ij} > \bar{p}_{ij}, \\ -1, & p_{ij} < \bar{p}_{ij}, \end{cases} \quad d_{ij} \in [-1, 1] \text{ if } p_{ij} = \bar{p}_{ij}.$$

**Insertion.** At the insertion boundary $p_{ij} = 0$ with continuous $\bar{p}_{ij} > 0$, we have $d_{ij} = -1$ if $\bar{p}_{ij} > 0$, but to capture the switch from 0 to 1 we consider the subgradient at the boundary:

$$p_{ij}^{(t+1)} = 1 \Longleftrightarrow 0 - \eta\left(g_{ij} + \beta\, d_{ij}\right) \leq -1 \Longleftrightarrow -g_{ij} > \beta\, d_{ij}.$$

Since at the 0-boundary we take $d_{ij} = +1$ for the "hardest" subgradient,

$$-g_{ij} > \beta$$

is required for insertion.

**Deletion.** At the deletion boundary $p_{ij} = 1$ with $\bar{p}_{ij} < 1$, we similarly take $d_{ij} = -1$ and find

$$p_{ij}^{(t+1)} = 0 \Longleftrightarrow 1 - \eta\left(g_{ij} + \beta\, d_{ij}\right) \geq 1 \Longleftrightarrow g_{ij} > \beta.$$

In both cases, an edge flips exactly when its *marginal benefit* or *cost* in the prediction loss exceeds the fixed penalty $\beta$, yielding a transparent sparsity threshold.

ROLE OF THE Γ MATRIX

Rather than using a uniform $\ell_1$ penalty $\beta \sum_{i,j} |\bar{P}_{ij} - A_{ij}|$, we shift the target by $\Gamma$ and write

$$\min_{\bar{P}} \ L_{\text{pred}}(\bar{P}) \ + \ \beta \sum_{i,j} |\bar{P}_{ij} - (A_{ij} + \gamma_{ij})|.$$

By simple algebraic manipulation, this is equivalent (up to an additive constant) to

$$\min_{\bar{P}} \ L_{\text{pred}}(\bar{P}) \ + \ \beta \sum_{i,j} w_{ij} |\bar{P}_{ij} - A_{ij}|, \quad w_{ij} \equiv \frac{1}{1 + \gamma_{ij}}.$$

In the latter form, the KKT subgradient for each $(i, j)$ is

$$\underbrace{\frac{\partial L_{\text{pred}}}{\partial \bar{P}_{ij}}}_{\text{model term}} + \beta \, w_{ij} \, \text{sign}(\bar{P}_{ij} - A_{ij}),$$

so a larger $\gamma_{ij} \Rightarrow$ smaller $w_{ij} \Rightarrow$ smaller magnitude of the regularizer $\Rightarrow$ cheaper to flip edge $(i, j)$, all **without** changing the global $\beta$.

## A.5 Smoothness of the Prediction Loss

We next show, at a high level, that our prediction loss $L_{\text{pred}}(P, N)$ admits an $L$-Lipschitz continuous gradient (i.e. is $L$-smooth) in the perturbation variables $(P, N)$.

**Lemma 2** *Let $f_\theta$ be a GNN with fixed weights $\theta$, built from layers that are each Lipschitz continuous (e.g. linear transforms, neighbor-aggregation, and elementwise activations such as ReLU). Define*

$$L_{\text{pred}}(P, N) = L\big(f_\theta(A \odot P, \ X + N), \ y\big),$$

*where $L$ is a twice-differentiable loss (e.g. cross-entropy). Then there exists a constant $L > 0$ depending only on $\theta$, the GNN architecture, and L, such that for all $(P, N)$ and $(P', N')$,*

$$\big\| \nabla L_{\text{pred}}(P, N) - \nabla L_{\text{pred}}(P', N') \big\|$$
$$\leq L \, \big\| (P, N) - (P', N') \big\|.$$

*Hence $L_{\text{pred}}$ is L-smooth.*

**Proof 2 (High-Level)** *Note that each GNN layer can be written as the composition of:*

- *A linear map in $(P, N)$ (adjacency mask enters via $A \odot P$, features via $X + N$), which is Lipschitz.*

- *An elementwise activation (e.g. ReLU or smooth ReLU), which is 1-Lipschitz.*

- *A final differentiable loss L, whose gradient is Lipschitz in the model's output.*

*By the chain rule, the gradient of the overall map $(P, N) \mapsto L\big(f_\theta(\cdot), y\big)$ is Lipschitz, with constant*

$$L \ \leq \ L_L \prod_{\ell=1}^{L} L_{layer,\ell},$$

*where $L_L$ is the loss's Lipschitz constant and each $L_{layer,\ell}$ upper-bounds the Lipschitz constant of layer $\ell$.*

**Remark.** In practice, we do not require the exact value of $L$, only the existence of such a bound to invoke standard convergence results for projected gradient methods on smooth + non-smooth objectives.

## A.6 Choosing the Trade-off

In practice, $\beta$ controls sparsity versus fidelity. A grid search typically selects $\beta \in [0.1, 1]$ to yield few edits while guaranteeing $\Phi(G') \neq \Phi(G)$. The step size ($\alpha$ is $\eta$ in Algorithm 1) is as well selected via a grid search in $\{10^{-3}, 10^{-2}, 10^{-1}, 1\}$.

## A.7 COMPUTATIONAL COMPLEXITY.

Each iteration of Algorithm 1 performs:

- one forward+backward pass through the GNN, which on a graph with $n$ nodes, $|E|$ edges, hidden-dimensionality $d$ and feature-dimensionality $f$ takes

$$O\big(|E|\,d + n\,d\,f\big),$$

  since message-passing scales linearly in edges and feature multiplications scale in $n\,f \times d$;

- elementwise thresholding of the perturbation masks $P$ and $N$, costing

$$O\big(|E| + n\,f\big),$$

  as we only inspect each edge entry and each feature entry once. And the STE back-pass adds no asymptotic overhead.

Thus, the dominant per-iteration cost is

$$O\big(|E|\,d + n\,d\,f\big),$$

i.e., linear in the graph size under standard GNN architectures, ensuring the method scales to large graphs. Recall that $K$ total iterations are performed in Algorithm 1. Treating $d$ as a small constant, when the graph is dense $|E| \sim n^2$, the complexity reduces to $O\big(n^2\big)$.

## A.8 EXTENSION TO WEIGHTED AND DIRECTED GRAPHS

Our theoretical developments extend straightforwardly to weighted or directed graphs. For weighted graphs, replace the binary mask $P \in \{0,1\}^{n \times n}$ by a continuous mask $P \in [a,b]^{n \times n}$ (e.g. $[0,1]$), and substitute the unweighted $\ell_1$ distance

$$\sum_{i,j} |p_{ij} - \bar{p}_{ij}|$$

with a weighted version

$$\sum_{i,j} w_{ij}\,\big|p_{ij} - \bar{p}_{ij}\big|,$$

where $w_{ij} > 0$ can reflect edge-specific costs. All projected gradient and subgradient arguments carry over by projecting onto the box $[a,b]$ and using the weighted sign subgradient for the $\ell_1$ term. For directed graphs, simply treat $(i,j)$ and $(j,i)$ as distinct entries in $P$, and the same "flip when $|\partial L_{\mathrm{pred}}/\partial p_{ij}| > \beta\,w_{ij}$" threshold holds. Convergence to a KKT-stationary point and the $\ell_1$-minimality bound remain valid with these modifications, since they rely only on box-constraints and separable $\ell_1$ penalties.

## A.9 SEARCH-SPACE EXPRESSIVITY

By jointly parameterizing the adjacency perturbation $\bar{P} \in [0,1]^{n \times n}$ and feature perturbation $\bar{N} \in \mathbb{R}^{n \times f}$ (see Section 3 and algorithm 1), our gradient-based explainer can, in principle, reach *any* discrete graph–feature configuration (once binarized via sigmoid and thresholding). In practice, however, the non-convex loss landscape can trap plain gradient descent in a local minimum within the basin around the original graph, favoring small, local edits. To counteract this, we add a small amount of Gaussian noise to $\bar{P}$ at initialization. These random perturbations diversify the initial gradient directions, helping the optimizer "hop" across low-gradient regions and leading to a richer exploration of structurally distinct counterfactuals without sacrificing convergence.

Hence, even when the optimization converges, the explainer may fail to reach a counterfactual due to either local minima or an ill-conditioned decision boundary in the oracle – e.g., the true label region may be disjoint, vanishingly thin, or entirely absent near the input. This is reflected in our experiments: some generated counterfactuals either fail to flip the label or result in out-of-distribution inputs. These issues could be mitigated by improving the oracle's inductive bias and generalization capacity.

## B DATASETS

To evaluate our approach against state-of-the-art GCE methods, we conducted experiments on 18 datasets (13 real and 5 synthetic) that cover diverse domains and involve multi-class graph classification tasks. Table 7 and table 8 illustrate the characteristics of the datasets used in this paper for graph and node classification-explanation, respectively. Common information are reported in table 7.

Table 7: Graph explanation datasets characteristics.

| | TCR | TG | BAS | MUTAG | BZR | COX2 | AIDS | BBBP | ENZYMES | PROTEINS | Fingerprint | COLLAB | COLOR-3 | TRIANGLES | MSRC | DBLP | IMDB | TWITTER |
|---|---|---|---|---|---|---|---|---|---|---|---|---|---|---|---|---|---|---|
| Avg # of Nodes | 28 | 64 | 64 | 17.93 | 35.75 | 41.22 | 15.69 | 25.95 | 32.63 | 39.06 | 7.06 | 74.49 | 61.31 | 20.85 | 77.52 | 10.48 | 19.77 | 4.03 |
| Node Attr. | – | – | 1 | – | – | – | – | – | 18 | 29 | 2 | – | 4 | – | – | – | – | – |
| Avg # of Edges | 27.75 | 65.01 | 64.01 | 19.79 | 38.36 | 43.45 | 16.20 | 24.06 | 62.14 | 72.82 | 5.76 | 2457.78 | 91.03 | 32.74 | 198.32 | 19.65 | 96.53 | 4.98 |
| # of Graphs | 5000 | 5000 | 5000 | 188 | 405 | 467 | 2000 | 2039 | 600 | 1113 | 2149 | 5000 | 10500 | 45000 | 563 | 19456 | 1000 | 144033 |
| # of Classes | 2 | 2 | 2 | 2 | 2 | 2 | 2 | 2 | 6 | 2 | 15 | 3 | 11 | 10 | 20 | 2 | 2 | 2 |
| Motif | Cycle | Grid | House | – | – | – | – | – | – | – | – | – | Color | Triangle | – | – | – | – |
| Category | Synthetic | Synthetic | Synthetic | Molecular | Molecular | Molecular | Molecular | Molecular | Bioinf. | Bioinf. | Comp.Vis. | Social net. | Synthetic | Synthetic | Comp.Vis. | Social net. | Social net. | Social net. |
| Oracle Test. Acc. | 100.00% | 99.72% | 100.00% | 88.30% | 99.01% | 98.72% | 99.80% | 99.41% | 95.67% | 98.02% | 74.97% | 69.60% | 91.68% | 99.18% | 42.27% | 80.83% | 77.80% | 91.32% |

Table 8: Node explanation datasets characteristics.

| | BAS | BZR | ENZYMES | MSRC | TWITTER |
|---|---|---|---|---|---|
| # of Node labels | 2 | 10 | 3 | 22 | 1323 |
| Oracle Test. Acc. | 100.00% | 99.90% | 87.81% | 92.81% | 3.41% |

**AIDS** (Riesen & Bunke, 2008) consists of graphs representing molecular compounds. These graphs are derived from the AIDS Antiviral Screen Database of Active Compounds. This data set consists of two classes (active and inactive), which represent molecules with or without activity against HIV. The molecules are converted into graphs in a straightforward manner by representing atoms as nodes and the covalent bonds as edges. Nodes are labeled with the number of the corresponding chemical symbol and edges by the valence of the linkage. There are 2,000 elements in total (1,600 inactive elements and 400 active elements).

**BAShapes** (Ying et al., 2019) is a synthetic dataset consisting of a base graph and motifs connected on the base. The base graph is a Barabasi-Albert (BA) graph with house-shaped motif attached to it. The resulting graph is further perturbed by adding $0.1N$ random edges. Following the generation done in Lucic et al. (2022), there are 8 nodes on the base graph with 5 edges connecting them. Each base graph has 7 motives connected to it.

**BBBP** (Blood-Brain Barrier Penetration) (Martins et al., 2012) is a dataset widely used in drug discovery and neurological research to develop machine learning models that predict blood-brain barrier permeability. The blood-brain barrier is a protective membrane that shields the central nervous system by regulating the passage of solutes. Its presence is a critical consideration in drug development, whether for designing molecules that target the central nervous system or for identifying compounds that should be restricted from crossing the barrier. BBBP contains binary labels for 2,053 curated molecules, indicating whether a compound can penetrate the blood-brain barrier. Specifically, 1,570 molecules can penetrate the barrier, while 483 cannot.

**BZR** and **COX2** (Sutherland et al., 2003) have different molecular compounds, 467 cyclooxygenase-2 (COX-2) inhibitors and 405 benzodiazepine receptor (BZR) ligands, respectively. These datasets are widely used in quantitative structure-activity relationships (QSAR) studies, which attempt to correlate the biological activities of compounds with their structural attributes, to help elucidate the mechanism by which they act and to predict the activities of novel derivatives.

**COLLAB** (Yanardag & Vishwanathan, 2015) is a social network dataset comprising 5000 scientific collaborations derived from 3 public collaboration datasets (Leskovec et al., 2005), namely, High Energy Physics, Condensed Matter Physics and Astro Physics. Ego-networks of different researchers from each field were generated, and each graph was labeled as the field of the researcher. The task is then to determine whether the egocollaboration graph of a researcher belongs to High Energy, Condensed Matter or Astro Physics field.

**COLORS-3** (Knyazev et al., 2019) is a synthetic dataset consisting of 10500 random graphs with 11 classes where features of each node are assigned to one of the three colors (red, green or blue), $\mathbf{p} \in \mathbb{R}^3$. The dataset is extended to higher $n$-dimensional cases $\mathbf{p} \in \mathbb{R}^n$.

**DBLP** (Pan et al., 2013) dataset consists of bibliography data in computer science. Each record is associated with a number of attributes such as abstract, authors, year, venue, title, and reference ID.

The graph stream is built by selecting the list of conferences and using the papers published in these conferences (in chronological order) to form a binary-class graph stream. The classification task is to predict whether a paper belongs to DBDM (database and data mining) or CVPR (computer vision and pattern recognition) field, by using the references and the title of each paper.

**ENZYMES** (Borgwardt et al., 2005) is a bio-informatics dataset consisting of a dataset of 600 enzymes, constructed from the Protein Data Bank (Berman et al., 2000) and labeled with their corresponding enzyme class labels from the BRENDA enzyme database (Schomburg et al., 2004). It includes 100 proteins from each of six classes (EC 1, EC 2, EC 3, EC 4, EC 5, EC 6), which represent proteins out of the six enzyme commission top level hierarchy (EC classes). The proteins are converted into graphs by representing the secondary structure elements of a protein with nodes and edges of an attributed graph. Nodes are labeled with their type (helix, sheet, or loop) and their amino acid sequence. Every node is connected with an edge to its three nearest neighbors in space. Edges are labeled with their type and the distance they represent in angstroms.

**Fingerprint** (Riesen & Bunke, 2008) is a computer vision dataset consisting of 2149 fingerprints that are converted into graphs by filtering the images and extracting regions that are relevant (Neuhaus & Bunke, 2005). In order to obtain graphs from fingerprint images, the relevant regions are binarized and a noise removal and thinning procedure is applied. This results in a skeletonized representation of the extracted regions. Ending points and bifurcation points of the skeletonized regions are represented by nodes. Additional nodes are inserted in regular intervals between ending points and bifurcation points. Finally, undirected edges are inserted to link nodes that are directly connected through a ridge in the skeleton. Each node is labeled with a two-dimensional attribute giving its position. The edges are attributed with an angle denoting the orientation of the edge with respect to the horizontal direction.

**IMDB** is a movie collaboration dataset in which actors/actresses and genre information of different movies on IMDB are collected. For each graph, nodes represent actors/actresses, and there is an edge between them if they appear in the same movie. Collaboration graphs on Action and Romance genres were generated and ego-networks for each actor/actress derived. Note that a movie can belong to both genres at the same time, therefore movies from Romance genre already included to the Action genre were discarded. Similarly to COLLAB dataset, each ego-network was labeled with the genre graph to which belongs to. The task is then simply to identify which genre an ego-network graph belongs to.

**MSRC21** (Neumann et al., 2016) is derived from the a state-of-the-art dataset in semantic image processing originally introduced by Winn et al. (2005). Each image is represented by a conditional Markov random field graph. The nodes of each graph are derived by oversegmenting the images using the quick shift algorithm (Vedaldi & Soatto, 2008), resulting in one graph among the superpixels of each image. Nodes are connected if the superpixels are adjacent, and each node can further be annotated with a semantic label. Semantic (ground-truth) node labels are derived by taking the mode ground-truth label of all pixels in the corresponding superpixel. Semantic labels are objects name plus a label void to handle objects that do not fall into one of the given classes. Images consisting of solely one semantic label were removed, leading to a classification task among 20 classes for MSRC21 dataset.

**MUTAG** (Debnath et al., 1991) is a widely used dataset consisting of 188 nitroaromatic chemical compounds divided into two classes according to their mutagenic effect on a Salmonella typhimurium bacterium. Input graphs represent chemical compounds, vertices stand for atoms and are labeled by the atom type (represented by one-hot encoding), while edges between vertices represent bonds between the corresponding atoms.

**PROTEINS** (Borgwardt et al., 2005) is a bio-informatics dataset comprising 1113 proteins from the dataset of enzymes (59%) and non-enzymes (41%) (Dobson & Doig, 2003). Proteins are modeled as feature vectors which indicate for each amino acid its fraction among all residues, its fraction of the surface area, the existence of ligands, the size of the largest surface pocket and the number of disulphide bonds.

**Tree-Cycles (TCR)** (Ying et al., 2019) is an emblematic synthetic dataset. Each instance constitutes a graph comprising a central tree motif and multiple cycle motifs connected through singular edges. The dataset encompasses two distinct classes: i.e., one for graphs without cycles (0) and another for

graphs containing cycles (1). The TC also allows control of the number of nodes, the number of cycles, and the number of nodes in them.

**Tree-Grid (TG)** is a synthetic dataset similar to *Tree-Cyles*, in which n-by-n grid motifs are attached to the main tree motif in place of cycle motifs. We used 5000 graphs with 64 nodes and randomly attached a 3-by-3 grid.

**TRIANGLES** (Knyazev et al., 2019) is a synthetic dataset comprising 45000 graphs. The task is to count the number of triangles in the graph, node degree features as one-hot vectors are added to all graphs, so that the oracle model can exploit both graph structure and features.

**TWITTER** (Pan et al., 2014)(Pan et al., 2015) dataset is extracted from twitter sentiment classification. Because of the inherently short and sparse nature, twitter sentiment analysis (i.e., predicting whether a tweet reflects a positive or a negative feeling) is a difficult task. To build a graph dataset, each tweet is represented as a graph by using tweet content, with nodes in each graph denoting the terms and/or smiley symbols and edges indicating the co-occurrence relationship between two words or symbols in each tweet. To ensure the quality of the graph, only tweets containing 20 or more words were used. Tweets from April 6 to June 16 were selected to generate 140,949 graphs (in a chronological order).

## C  DETAILED EXPERIMENTAL SETUP

### C.1  ORACLES AND TRAINING

The oracle Graph Convolutional Network (GCN) models for different datasets were trained with varying architectures and hyperparameters, all using RMSprop optimization and CrossEntropyLoss, with an 80%-20% train-test split. Most datasets (BAS, AIDS, BZR, COX2, TCR, TG, and EN-ZYMES) used 3 convolutional layers and 1 dense layer, BBBP (5 conv, 3 dense), COX2, MUTAG (2 conv, 2 dense), TRIANGLES (5 conv, 2 dense), COLORS-3 (2 conv, 1 dense), COLLAB, PRO-TEINS (3 conv, 2 dense), Fingerprint (5 conv, 5 dense) . Learning rates varied between 0.001 and 0.01, with training epochs ranging from 20 to 1000, and batch sizes typically 32, except BBBP, ENZYMES, PROTEINS (64), Fingerprint (128), COLLAB (256), TRIANGLES (1024). The experiments were conducted on an 8GB NVIDIA RTX 4060 GPU with an Intel Core i9-13900HX and 32GB RAM, while node explanation experiments used a 4GB NVIDIA RTX 3050 GPU with an Intel Core i7-11800H and 16GB RAM.

### C.2  HYPERPARAMETERS SEARCH

We perform a hyperparameter search according to these combination of parameters: $\alpha \in \{0.001, 0.01, 0.1, 1\}, \beta \in \{0, 0.5, 1\}, K \in \{50, 100, 500, 5000\}, \gamma \in \{0, 0.1, 0.01, 0.001\}$. The optimal configuration found was $\alpha = 0.1, \beta = 0.5, K = 50$ and $\gamma = 0$; if accuracy is preferred over other metrics (see Section 4.4), then $\gamma = 0.01$ performed best over the other choices.

### C.3  METRICS

**Oracle Accuracy** tells us how accurate the model $\Phi$ is at predicting the ground truth labels, $\frac{1}{N}\sum_{i=1}^{N}\mathbf{1}[\Phi(G_i) = y_i]$.

**Validity** illustrates the capability of the explainer to cross the decision boundary of $\Phi$ given an input $G_i$, $\frac{1}{N}\sum_{i=1}^{N}\mathbf{1}[\Phi(G_i') \neq \Phi(G_i)]$.

**Fidelity** Fidelity measures how faithful the explainer's generated counterfactuals are to the original instance's true labels, not just to the oracle's learned decision boundary. $\frac{1}{N}\sum_{i=1}^{N}\max(\mathbf{1}(\Phi(G_i) = y_i) - \mathbf{1}[\Phi(G_i') = y_i], 0)$.

Note that if the oracle misclassifies an instance, the explainer may produce a counterfactual that changes the predicted class to the original class; in such cases, accuracy would indicate success even though the counterfactual is not a true solution, whereas fidelity correctly reflects this.

**Sparsity** measures the ratio between the number of structural features modified to obtain a valid counterfactual explanation and the number of structural features in the original instance.

**Graph Edit Distance (GED)** tracks the modification applied to the graph of the valid counterfactual explanation, i.e., number of added/removed nodes/edges.

**Oracle Calls** is the number of times the explainer calls $\Phi$ to generate a valid counterfactual explanation.

**Runtime to counterfactual**: the time in seconds required by the explainer to generate a valid counterfactual explanations.

Table 9 shows all the performances of XPlore against the SoTA methods. Figure 4 shows the runtime of all explainers, showcasing that XPlore is lightweight and only second to CLEAR and $CF^2$. However, it shows that XPlore is more efficient than its baseline CF-GNNExpl. which is, notice, more restrictive in perturbating the input graph (i.e., only allowing edge deletions). Contrarily, XPlore allows for both edge additions and removals as well as node perturbations. Figure 5 presents the CS and GED distributions, showing that XPlore retains high semantic meaning in the CFs it generates. XPlore performs better in that it can also identify CFs for hard instances where competitors fail. Consequently, it achieves CS scores that are comparable to or better than those of other baselines on easy samples, while maintaining strong semantic meaning for hard instances not captured by competitors.

### C.4 OUT-OF-DISTRIBUTION EFFECT ANALYSIS

**Comparison of XPlore and RSGG: t-SNE projection of Wavelet Characteristic embeddings (Tree-Cycle dataset)** We can compare behaviours of XPlore against the second best explainer over the Tree-Cycle dataset. Depending on the original instance, we can see distinct behaviours of the two explainers taking place, let's analyze them.

XPLORE UNABLE TO LAND ON TREE DISTRIBUTION FOR A CYCLE CF XPlore sometimes is unable to land on Tree distribution for a Cycle CF (6a). 6b is a significative error by XPlore. RSGG is sometimes unable to correctly land on the Tree distribution as well (6c).

XPLORE STREGHTS, RSGG WEAKNESSES Looking at counterfactuals for the Tree class, the results show an **opposite trend**, XPlore is able to **correctly land CFs** on the Cycle distribution, while RSGG struggles to generate in-distribution CFs, not always landing on the Cycle distribution: (6d); yet sometimes, both succeed (6e).

**Comparison of XPlore, CFGNNE and RSGG: t-SNE projection of Wavelet Characteristic embeddings (Tree-Cycle dataset)** We now compare togheter CFs for XPlore, CFGNNE and RSGG. Here we show only CFs for the Tree class, as we plot only instances where all 3 explainers were able to find a counterfactual, and CFGNNE has success only in finding CFs for this class.

CFGNNE DOES NOT LAND ON CYCLE DISTRIBUTION CFGNNE does not land on Cycle distribution (7a), this is expected as **CFGNNE only removes edges**, hence it cannot produce Cycles. Note how **XPlore and RSGG behave similarly**.
Sometimes, **XPlore is the only explainer finding a solution** (7b).
Rarely, the embedder gets tricked (we know CFGNNE cannot produce cycles) (7c).

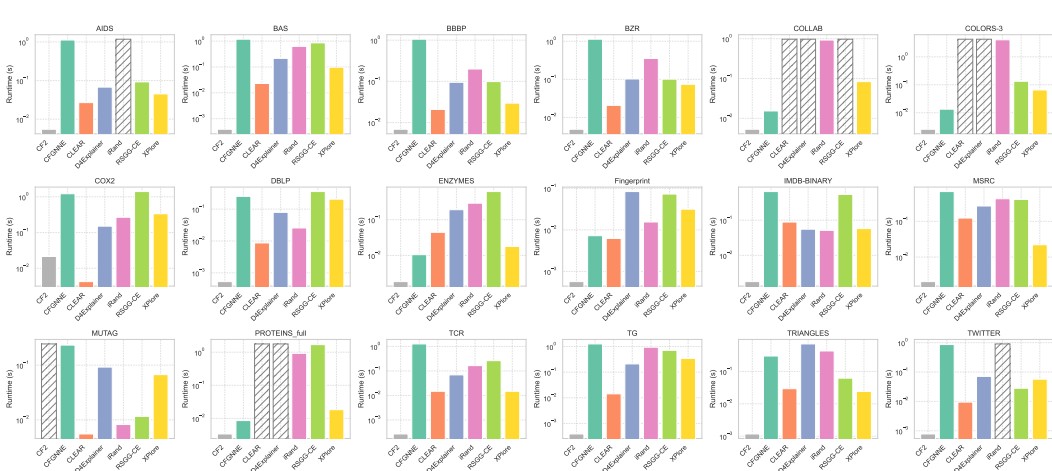

Figure 4: Runtime (s) at inference across multiple datasets. **XPlore (yellow) maintains a competitive runtime, making it the most reliable explainer in practice.** Placeholders are used for explainers with 0% accuracy.

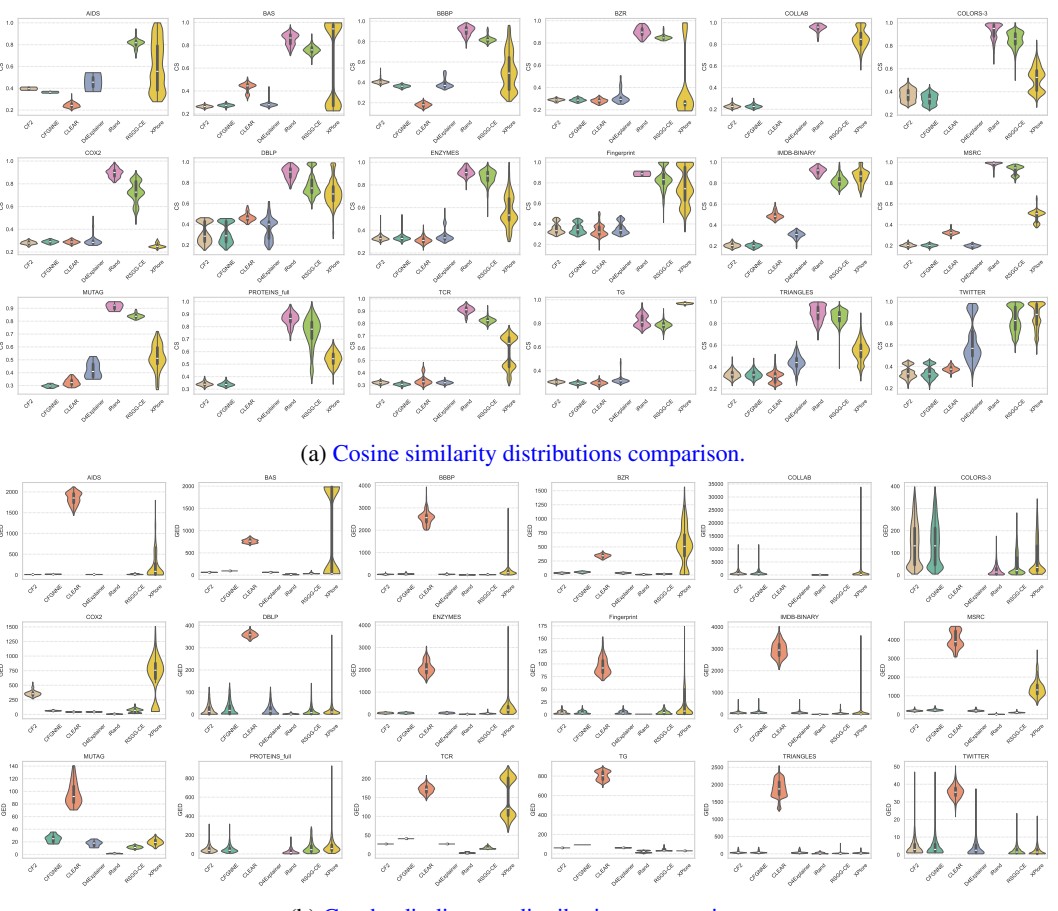

(a) Cosine similarity distributions comparison.

(b) Graph edit distance distributions comparison.

Figure 5: XPlore identifies more CFs, thus capturing harder instances. Elongated and bimodal distributions reflect competitive performance and semantic meaning retention.

Table 9: Extended comparison of XPlore with SoTA methods on the used datasets. Standard deviation is reported. Bold is best-performing; underline is second-best. Highlighted columns depict the most important metrics as they evaluate the faithfulness of the explainer to produce counterfactuals. When an explainer cannot produce valid counterfactuals in a dataset, it does not make sense to evaluate the other metrics (see −).

| Dataset | Method | Validity↑ | Fidelity↑ | Sparsity↓ | Oracle Calls↓ |
|---|---|---|---|---|---|
| TCR | iRand | 27.92% ± 44.86% | 0.279 ± 0.449 | **0.062** ± 0.023 | 10.666 ± 0.084 |
| | CF² | 50.04% ± 50.00% | 0.500 ± 0.500 | 0.491 ± 0.000 | **0.000** ± 0.000 |
| | CLEAR | 50.68% ± 50.00% | 0.507 ± 0.500 | 3.037 ± 0.193 | **0.000** ± 0.004 |
| | RSGG-CE | 67.90% ± 46.69% | 0.679 ± 0.467 | 0.305 ± 0.057 | 8.890 ± 0.169 |
| | D4Explainer | 44.82% ± 49.73% | 0.448 ± 0.497 | 0.491 ± 0.000 | **0.000** 0.012 |
| | CF-GNNExpl | 50.04% ± 50.00% | 0.500 ± 0.500 | 0.745 ± 0.000 | 46.000 ± 0.399 |
| | XPlore | **100.00%** ± 0.00% | **1.000** ± 0.000 | 2.708 ± 0.906 | 3.593 ± 0.009 |
| BAS | iRand | 50.70% ± 50.00% | 0.507 ± 0.500 | **0.112** ± 0.044 | 43.600 ± 0.293 |
| | CF² | 45.78% ± 49.82% | 0.457 ± 0.499 | 0.496 ± 0.000 | **0.000** ± 0.001 |
| | CLEAR | 50.96% ± 49.99% | 0.510 ± 0.500 | 6.007 ± 0.325 | **0.000** ± 0.035 |
| | RSGG-CE | 91.04% ± 28.56% | 0.910 ± 0.286 | 0.331 ± 0.108 | 23.585 ± 1.069 |
| | D4Explainer | 44.56% ± 49.70% | 0.446 ± 0.497 | 0.499 ± 0.002 | **0.000** ± 0.014 |
| | CF-GNNExpl | 44.18% ± 49.66% | 0.442 ± 0.497 | 0.748 ± 0.000 | 46.000 ± 0.283 |
| | XPlore | **100.00%** ± 0.00% | **1.000** ± 0.000 | 7.600 ± 7.557 | 2.514 ± 0.047 |
| BZR | iRand | 27.16% ± 44.48% | 0.262 ± 0.451 | **0.104** ± 0.045 | 25.373 ± 0.213 |
| | CF² | 19.75% ± 39.81% | 0.188 ± 0.403 | 0.516 ± 0.007 | **0.000** ± 0.002 |
| | CLEAR | 60.49% ± 48.89% | 0.595 ± 0.501 | 4.937 ± 0.843 | **0.000** ± 0.012 |
| | RSGG-CE | 21.23% ± 40.90% | 0.202 ± 0.414 | 0.258 ± 0.003 | 2.000 ± 0.015 |
| | D4Explainer | 20.00% ± 40.00% | 0.190 ± 0.405 | 0.523 ± 0.006 | **0.000** ± 0.015 |
| | CF-GNNExpl | 19.75% ± 39.81% | 0.188 ± 0.403 | 0.758 ± 0.003 | 46.000 ± 0.224 |
| | XPlore | **100.00%** ± 0.00% | **0.980** ± 0.198 | 6.595 ± 3.616 | 2.244 ± 0.029 |
| AIDS | iRand | 0.00% ± 0.00% | 0.000 ± 0.000 | – | – |
| | CF² | 0.10% ± 3.16% | 0.001 ± 0.032 | 3.870 ± 0.011 | **0.000** ± 0.004 |
| | CLEAR | 16.75% ± 37.34% | 0.168 ± 0.373 | 13.442 ± 1.834 | **0.000** ± 0.008 |
| | RSGG-CE | 19.80% ± 39.85% | 0.198 ± 0.398 | **0.258** ± 0.004 | 2.000 ± 0.027 |
| | D4Explainer | 0.10% ± 0.032% | 0.001 ± 0.032 | 0.488 ± 0.031 | **0.000** ± 0.001 |
| | CF-GNNExpl | 0.10% ± 3.16% | 0.001 ± 0.032 | 0.745 ± 0.005 | 46.000 ± 0.059 |
| | XPlore | **32.30%** ± 45.96% | **0.323** ± 0.460 | 2.445 ± 2.002 | 5.851± 0.058 |
| ENZYMES | iRand | 26.67% ± 44.22% | 0.238 ± 0.445 | **0.074** ± 0.046 | 29.069 ± 0.046 |
| | CF² | 68.33% ± 46.52% | 0.633 ± 0.502 | 0.657 ± 0.034 | **0.000** ± 0.004 |
| | CLEAR | 83.17% ± 37.42% | 0.797 0.402 | 27.942 ± 24.021 | **0.000** ± 0.017 |
| | RSGG-CE | 98.33% ± 12.80% | 0.930 ± 0.292% | 0.396 ± 0.152 | 19.400 ± 2.801 |
| | D4Explainer | 68.00% ± 46.65% | 0.632 ± 0.499 | 0.663 ± 0.499 | **0.000** ± 0.066 |
| | CF-GNNExpl | 68.33% ± 46.52% | 0.633 ± 0.502 | 0.657 ± 0.034 | 2.000 ± 0.002 |
| | XPlore | **100.00%** ± 0.00% | **0.942** ± 0.291 | 2.394 ± 1.538 | 2.615 ± |
| Fingerprint | iRand | 0.09% ± 3.05% | 0.001 ± 0.030% | **0.031** ± 0.000 | 3.500 ± 0.007 |
| | CF² | 24.52% ± 43.02% | 0.133 ± 0.400 | 0.412 ± 0.052 | **0.000** ± 0.000 |
| | CLEAR | 72.73% ± 44.53% | 0.515 ± 0.537 | 15.517 ± 14.400 | **0.000** ± 0.001 |
| | RSGG-CE | 90.46% ± 29.38% | 0.653 ± 0.566 | 0.329 ± 0.142 | 6.479 ± 0.048 |
| | D4Explainer | 24.52% ± 43.02% | 0.133 ± 0.400 | 0.412 ± 0.052 | **0.000** ± 0.014 |
| | CF-GNNExpl | 24.52% ± 43.02% | 0.133 ± 0.400 | 0.412 ± 0.052 | 2.000 ± 0.002 |
| | XPlore | **100.00%** ± 0.00% | **0.730** ± 0.487 | 0.872 ± 0.591 | 4.168 ± 0.052 |
| COLLAB | iRand | 4.60% ± 20.95% | 0.006 ± 0.213 | **0.018** ± 0.021 | 58.696 ± 2.505 |
| | CF² | 52.66% ± 49.93% | 0.262 ± 0.624 | 0.886 ± 0.061 | **0.000** ± 0.003 |
| | CLEAR | 0.00% ± 0.00% | 0.000 ± 0.000 | – | – |
| | RSGG-CE | 0.00% ± 0.00% | 0.000 ± 0.000 | – | – |
| | D4Explainer | 0.00% ± 0.00% | 0.000 ± 0.000 | – | – |
| | CF-GNNExpl | 52.66% ± 49.93% | 0.262 ± 0.624 | 0.886 ± 0.061 | 2.000 ± 0.004 |
| | XPlore | **100.00%** ± 0.00% | **0.570** ± 0.709 | 0.810 ± 0.068 | 3.604 ± 0.021 |
| IMDB | iRand | 3.70% ± 18.88% | 0.026 ± 0.192 | **0.024** ± 0.013 | 10.865 ± 0.048 |
| | CF² | 50.60% ± 50.00% | 0.366 ± 0.674 | 0.803 ± 0.038 | **0.000** ± 0.001 |
| | CLEAR | 56.20% ± 49.61% | 0.420 ± 0.696 | 34.773 ± 15.650 | **0.000** ± 0.022 |
| | RSGG-CE | 86.10% ± 34.60% | **0.664** ± 0.802 | 0.462 ± 0.122 | 34.890 ± 1.795 |
| | D4Explainer | 49.60% ± 50.00% | 0.358 ± 0.669 | 0.797 ± 0.037 | **0.000** ± 0.015 |
| | CF-GNNExpl | 50.60% ± 50.00% | 0.366 ± 0.674 | 0.902 ± 0.019 | 46.000 ± 0.230 |
| | XPlore | 78.80 ± 40.87% | 0.606 ± 0.780 | 0.390 ± 0.068 | 5.042 ± 0.047 |
| COLORS-3 | iRand | 42.99% ± 49.51% | 0.300 ± 0.571 | **0.076** ± 0.064 | 66.175 ± 62.151 |
| | CF² | 52.07% ± 49.96% | 0.398 ± 0.568 | 0.598 ± 0.046 | **0.000** ± 0.001 |
| | CLEAR | 0.00% ± 0.00% | 0.000 ± 0.000 | – | – |
| | RSGG-CE | 94.57% ± 22.66% | 0.824 ± 0.471 | 0.312 ± 0.079 | 4.040 ± 0.848 |
| | D4Explainer | 0.00% ± 0.00% | 0.000 ± 0.000 | – | – |
| | CF-GNNExpl | 52.07% ± 49.96% | 0.398 ± 0.568 | 0.598 ± 0.046 | 2.000 ±0.007 |
| | XPlore | **100.00%** ± 0.00% | **0.871** ± 0.453 | 0.536 ± 0.057 | 12.204 ± 0.155 |

| Dataset | Method | Validity↑ | Fidelity↑ | Sparsity↓ | Oracle Calls↓ |
|---|---|---|---|---|---|
| TG | iRand | 36.16% ± 48.05% | 0.356 ± 0.485 | **0.175** ± 0.086 | 69.177 ± 0.0542 |
| | CF² | 49.86% ± 50.00% | 0.499 ± 0.500 | 0.496 ± 0.000 | **0.000** ± 0.001 |
| | CLEAR | 58.40% ± 49.29% | 0.584 ± 0.493 | 6.353 ± 0.400 | **0.000** ± 0.003 |
| | RSGG-CE | 89.28% ± 30.94% | 0.888 ± 0.324 | 0.316 ± 0.091 | 19.152 ± 0.911 |
| | D4Explainer | 49.86% ± 50.00% | 0.499 ± 0.500 | 0.500 ± 0.001 | **0.000** ± 0.013 |
| | CF-GNNExpl | 49.86% ± 50.00% | 0.499 ± 0.500 | 0.784 ± 0.000 | 46.000 ± 0.414 |
| | XPlore | **100.00%** ± 0.00% | **0.994** ± 0.106 | 0.248 ± 0.004 | 14.692 ± 0.236 |
| MUTAG | iRand | 2.66% ± 16.09 | 0.005 ± 0.163 | **0.035** ± 0.011 | 4.200 ± 0.004 |
| | CF² | 0.00% ± 0.00% | 0.005 ± 0.318 | – | – |
| | CLEAR | 35.11% ± 47.73% | 0.309 ± 0.506 | 2.438 ± 0.375 | **0.000** ± 0.001 |
| | RSGG-CE | 56.91% ± 49.52% | 0.516 ± 0.550 | 0.264 ± 0.004 | 2.000 ± 0.002 |
| | D4Explainer | 9.57% ± 29.42% | 0.021 ± 0.309 | 0.526 ± 0.016 | **0.000** ± 0.007 |
| | CF-GNNExpl | 10.11% ± 30.14% | 0.005 ± 0.318 | 0.758 ± 0.005 | 46.000 ± 0.147 |
| | XPlore | **67.55%** ± 46.82% | **0.548** ± 0.613 | 0.459 ± 0.036 | 2.866 ± 0.089 |
| COX2 | iRand | 69.81% ± 45.91% | 0.677 ± 0.490 | **0.087** ± 0.049 | 22.482 ± 0.192 |
| | CF² | 24.20% ± 42.83% | 0.229 ± 0.435 | 4.354 ± 0.235 | **0.000** ± 0.005 |
| | CLEAR | 22.06% ± 41.46% | 0.208 ± 0.421 | 0.512 ± 0.001 | **0.000** ± 0.002 |
| | RSGG-CE | **99.36%** ± 7.99% | **0.968** ± 0.238 | 0.761 0.366 | 92.914 ± 1.102 |
| | D4Explainer | 22.06% ± 41.46% | 0.208 ± 0.421 | 0.518 ± 0.003 | **0.000** ± 0.025 |
| | CF-GNNExpl | 22.06% ± 41.46% | 0.208 ± 0.421 | 0.756 ± 0.001 | 46.000 0.198 |
| | XPlore | **99.36%** ± 7.99% | **0.968** ± 0.238 | 7.682 3.704 | 11.388 ± 0.568 |
| BBBP | iRand | 19.76% ± 39.82% | 0.275 ± 0.460 | **0.059** ± 0.037 | 12.928 ± 0.190 |
| | CF² | 25.26% ± 43.45% | 0.253 ± 0.434 | 0.510 ± 0.024 | **0.000** ± 0.002 |
| | CLEAR | 22.90% ± 42.02% | 0.229 ± 0.420 | 51.889 ± 32.182 | **0.000** ± 0.022 |
| | RSGG-CE | 22.90% ± 42.02% | 0.229 ± 0.420 | 0.258 ± 0.006 | 2.000 8.131 |
| | D4Explainer | 21.24% ± 40.90% | 0.212 ± 0.409 | 0.524 ± 0.010 | **0.000** ± 0.038 |
| | CF-GNNExpl | 22.31% ± 41.64% | 0.223 ± 0.416 | 0.758 ± 0.005 | 46.000 ± 0.185 |
| | XPlore | **81.51%** % 38.82% | **0.803** ± 0.412 | 2.217 ± 1.206 | 4.055 ± 0.030 |
| PROTEINS | iRand | 18.87% ± 39.13% | 0.183 ± 0.394 | **0.107** ± 0.078 | 75.524 ± 4.036 |
| | CF² | 16.35% ± 36.98% | 0.151 ± 0.375 | 0.651 ± 0.025 | **0.000** ± 0.001 |
| | CLEAR | 0.00% ± 0.000% | 0.000 ± 0.000 | – | – |
| | RSGG-CE | 58.67% 49.24% | 0.556 ± 0.527 | 0.664 ± 0.477 | 78.757 ± 3.667 |
| | D4Explainer | 0.00% ± 0.00% | 0.000 ± 0.000 | – | – |
| | CF-GNNExpl | 16.35% ± 36.98% | 0.151 ± 0.375 | 0.651 ± 0.025 | 2.000 ± 0.002 |
| | XPlore | **65.41%** ± 47.57% | **0.627** ± 0.511 | 0.569 ± 0.039 | 2.798 ± 0.006 |
| MSRC | iRand | 95.74% ± 20.20% | 0.391 ± 0.656 | **0.027** ± 0.039 | 24.731 ± 0.383 |
| | CF² | 90.94% ± 28.70% | 0.371 ± 0.523 | 0.719 ± 0.009 | **0.000** ± 0.000 |
| | CLEAR | 23.98% ± 42.70% | 0.012 ± 0.158 | 10.630 ± 1.095 | **0.000** ± 0.034 |
| | RSGG-CE | **100.00%** ± 0.00% | 0.423 ± 0.538 | 0.368 ± 0.050 | 7.536 ± 1.666 |
| | D4Explainer | 90.94% ± 28.70% | 0.371 ± 0.523 | 0.721 ± 0.009 | **0.000** ± 0.047 |
| | CF-GNNExpl | 90.94% ± 28.70% | 0.371 ± 0.523 | 0.859 ± 0.004 | 46.000 ± 0.078 |
| | XPlore | **100.00%** ± 0.00% | **0.423** ± 0.569 | 5.059 ± 0.756 | 2.089 ± 0.006 |
| DBLP | iRand | 1.17% ± 10.74% | 0.006 ± 0.108 | **0.032** ± 0.019 | 8.780 ± 0.018 |
| | CF² | 5.76% ± 23.29% | 0.027 ± 0.240 | 0.530 ± 0.158 | **0.000** ± 0.001 |
| | CLEAR | 0.68% ± 8.21% | 0.005 ± 0.082 | 3.528 ± 1.037 | **0.000** ± 0.003 |
| | RSGG-CE | 57.51% ± 49.43% | 0.450 ± 0.686 | 0.419 ± 0.225 | 21.848 ± 0.844 |
| | D4Explainer | 7.02% ± 25.55% | 0.031 ± 0.265 | 0.599 ± 0.089 | **0.000** ± 0.021 |
| | CF-GNNExpl | 5.75% ± 23.29% | 0.027 ± 0.240 | 0.765 ± 0.079 | 46.000 ± 0.035 |
| | XPlore | **91.84%** ± 27.37% | **0.748** ± 0.765 | 0.569 ± 0.278 | 36.115 ± 0.097 |
| TWITTER | iRand | 0.00% ± 0.00% | 0.000 ± 0.000 | – | – |
| | CF² | 38.82% ± 48.74% | 0.345 ± 0.545 | 0.476 ± 0.111 | **0.000** ± 0.001 |
| | CLEAR | 7.65% ± 26.58% | 0.039 ± 0.277 | 2.352 ± 0.246 | **0.000** ± 0.003 |
| | RSGG-CE | 48.36% ± 49.97% | 0.434 ± 0.579 | 0.238 ± 0.056 | 2.000 ± 0.003 |
| | D4Explainer | 23.45% ± 42.37% | 0.161 ± 0.476 | 0.386 ± 0.157 | **0.000** ± 0.012 |
| | CF-GNNExpl | 38.82% ± 48.74% | 0.345 ± 0.545 | 0.476 ± 0.111 | 1.000 ± 0.061 |
| | XPlore | **50.71%** ± 50.00% | **0.613** ± 0.626 | 0.217 ± 0.143 | 1.000 ± 0.027 |
| TRIANGLES | iRand | 6.38% ± 24.44% | 0.053 ± 0.247 | **0.123** ± 0.080 | 71.421 ± 0.346 |
| | CF² | 37.13% ± 48.31% | 0.364 ± 0.488 | 0.621 0.032 | **0.000** ± 0.001 |
| | CLEAR | 89.99% ± 30.01% | 0.892 ± 0.311 | 53.517 ± 36.786 | **0.000** ± 0.019 |
| | RSGG-CE | 99.84% ± 4.00% | 0.987 ± 0.141 | 0.325 ± 0.080 | 5.161 ± 0.378 |
| | D4Explainer | 31.11% ± 46.29% | 0.302 ± 0.468 | 0.619 ± 0.038 | **0.000** ± 76.890 |
| | CF-GNNExpl | 37.13% ± 48.31% | 0.364 ± 0.488 | 0.621 ± 0.032 | 46.000 ± 0.056 |
| | XPlore | **100.00%** ± 0.00% | **0.988** ± 0.138 | 0.532 ± 0.053 | 3.366 ± 0.026 |

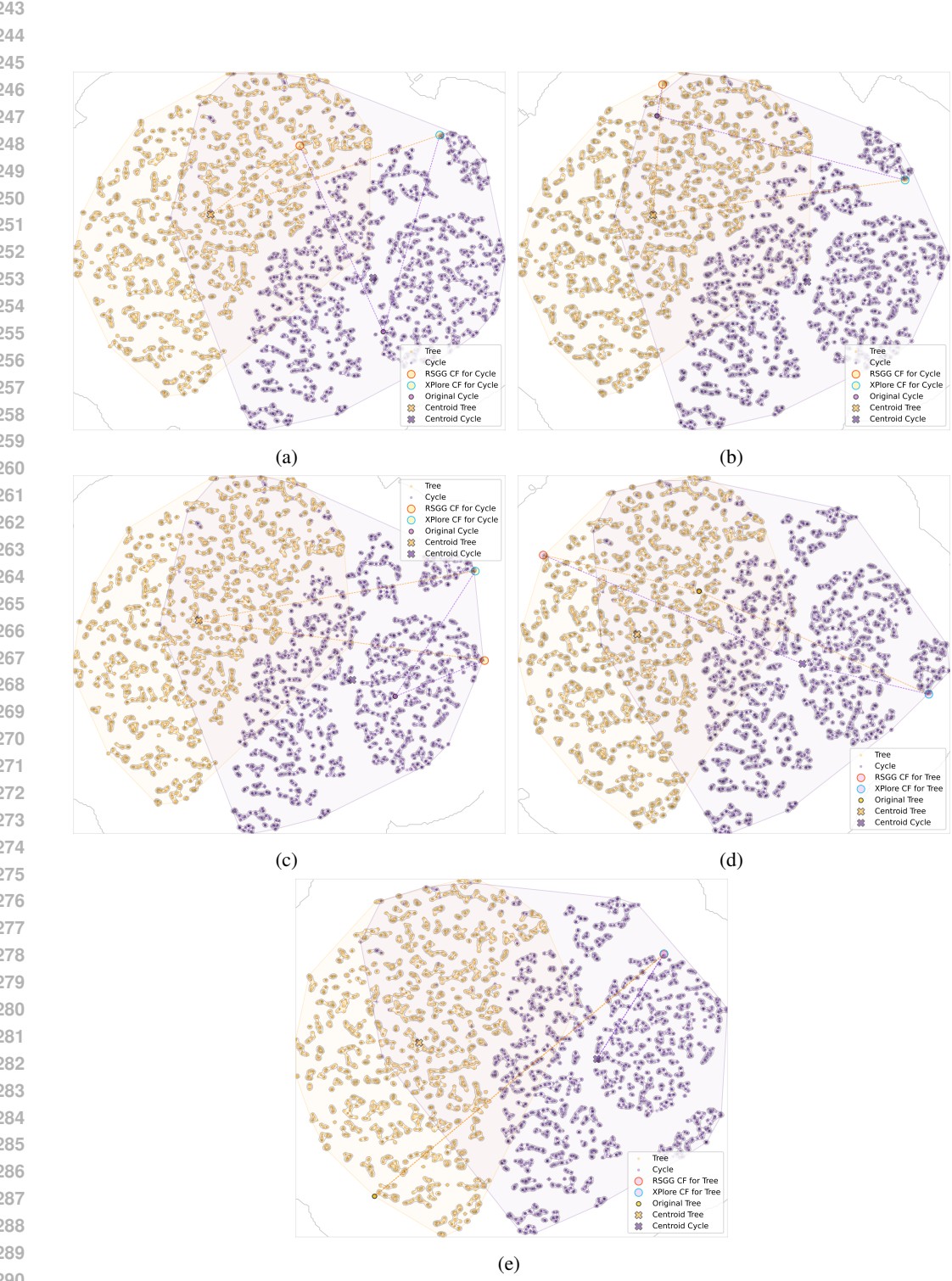

Figure 6: Comparison of XPlore and RSGG: t-SNE projection of Wavelet Characteristic embeddings (Tree-Cycle dataset)

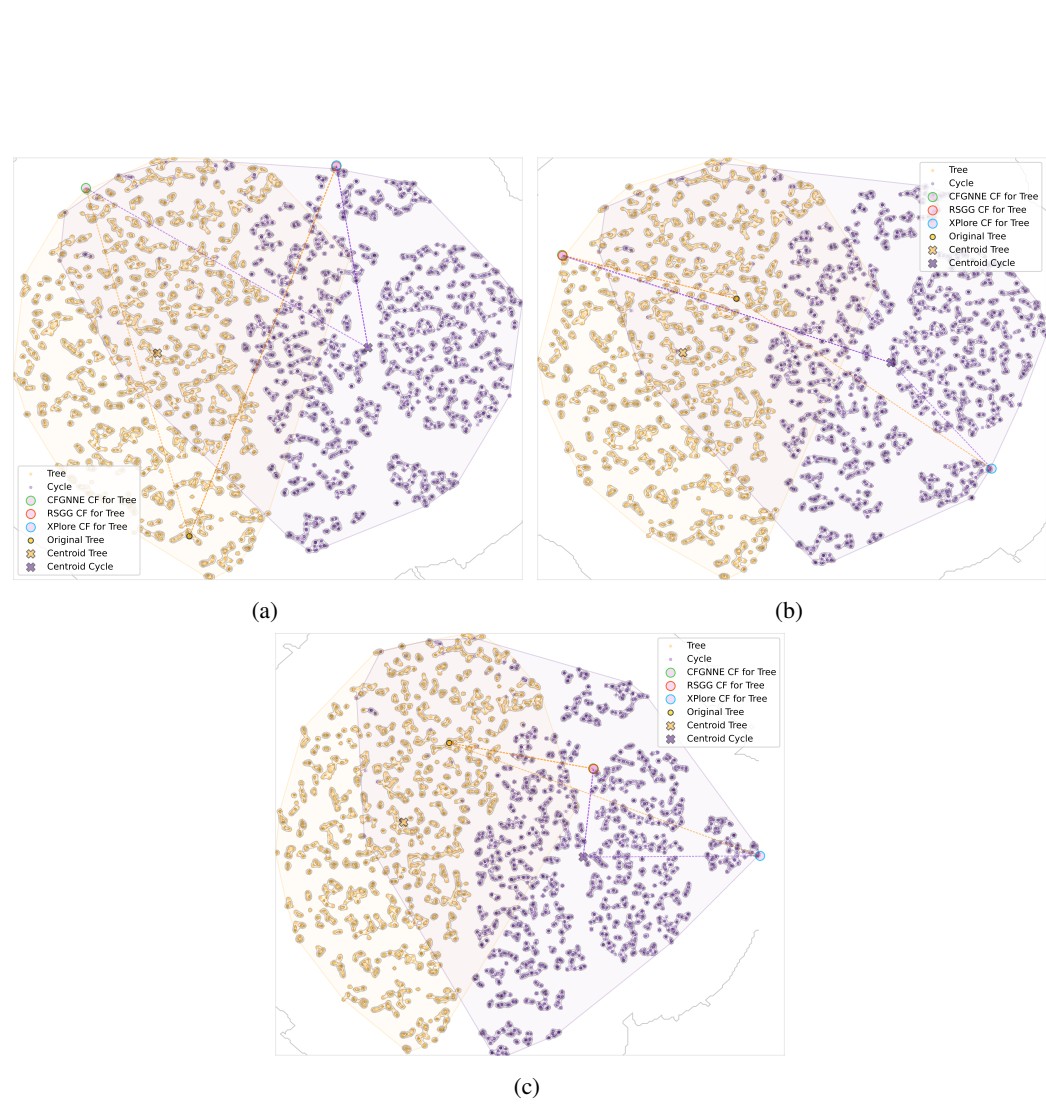

(a)

(b)

(c)

Figure 7: Comparison of XPlore, CFGNNE and RSGG: t-SNE projection of Wavelet Characteristic embeddings (Tree-Cycle dataset)

