# OpenReview forum: "Beyond Edge Deletion: A Comprehensive Approach to Counterfactual Explanation in Graph Neural Networks"
_ICLR.cc/2026/Conference — Submitted to ICLR 2026_

### Official Review · Reviewer_mmHe · 2025-10-28

**Soundness:** 1
**Presentation:** 2
**Contribution:** 2
**Rating:** 2
**Confidence:** 5

**Summary:**

The authors present Xplore, a counterfactual (CF) explanation method for graph neural networks. The method is derived from CF-GNNexplainer (Lucic et al.), but with a diffusion twist that allows for edge addition. Experiments are performed on different datasets and baselines for the problem of finding counterfactual.

**Strengths:**

- The overall idea (of gradient guided optimization) is interesting.
- The experiments consider a good amount of datasets.
- The mathematical proof of convergence is sound, although very standard and too slow.

**Weaknesses:**

- Presentation: Some parts of the paper are hard to read. The literature citations completely stop the flow of the text, Appendix section are poorly referenced. The paper does not seem to have been proofread with the applied ICLR format.

- In the contribution stated claim 2) seem incorrect, claim 1) should be reformulated.

- The literature review is outdated, and omits works published after early 2024. The paper only uses work from 2023 and before, and ignores relevant baselines from 2023, 2024, and 2025.

- The claim of novelty of adding edges or perturbing features is false, as seen in works such as: (1) Empowering Counterfactual Reasoning over Graph Neural Networks through Inductivity, Verma et al, 2023; (2) Global Counterfactual Explainer for Graph Neural Networks, Kosan et al, 2022; (3) COMBINEX: A Unified Counterfactual Explainer for Graph Neural Networks via Node Feature and Structural Perturbations, Giogi et al, 2025

- Following these gaps in the literature review, important recent baselines are missing.

- The analysis of the experimental results is extremely lacking: in CF explanation, there is a trade-off between validity of the counterfactual and its distance to the original graphs. See Lucic et al. (2022), CF-GNNExplainer, or Ma et al. (2022), CLEAR. This trade-off is nowhere mentioned, and absent from the analysis. The good validity results of the method may be entirely explained by the algorithm not stopping until it finds any valid counterfactual.

**Questions:**

- Part 1: claim 1 seems to express that you are the first to consider edge and node deletion, but this is not true, as for instance the cited Ma et al.’s CLEAR already does this.

- Part 1: claim 2 appears incorrect as well, ”the closest counterfactual through directed modifications” is not mentioned in the rest of the paper, and in fact, as seen in Table 3 for GED, seems widely incorrect.

- Part 2: Since D4Explainer also uses diffusion to find CF through denoising diffusion, how does your method differ? Please add a deeper comparison with this paper.

- Part 3.1: Equation 2 is awkwardly introduced, and does not serve any purpose; the loss used is given in Equation 7. You should introduce the metrics for your objective here, not the loss.

- Part 3.1: the Node Counterfactual Explanation is unclear, and should go after your method or be more general. It fails to explain what Node Counterfactual Explanation is. Please state the objective.

- Part 3.2: As mentioned, equation (3) is a subcase of Lucic et al.’s work where the subgraph considered is the whole graph.

- Part 3.2: The idea of noisy perturbation then denoising is interesting, and very similar to diffusion. I would reframe the work this way.

- Part 4: Table 3 is misleading, and does not relate to the objective stated: you should compare fidelity/validity and GED/CS at the same time for each method. Fidelity aims at finding counterfactuals, GED looks for good, i.e close counterfactual. Hence both should be analyzed together, as there is a trade-off.

- Part 4: Table 3 and validity. Getting 100 % validity is not surprising since the algorithm stops when it finds a counterfactual. The comparison with baselines seems unfair. This is also NOT discussed anywhere in the paper, which is a major issue.

- Part 4: unsurprisingly, the GED/CS of the proposed method is much higher than that of other methods, since the algorithm.

- Part 4.3: I am not sure what is the purpose of this part, this is not introduced or mentioned in the paper, and poorly structured.

- Appendix A.1: I am puzzled as to why you rewrote the proof of an already proven theorem. It is sufficient to just cite a theorem and use its result.

---

> ### Author Response · Authors · 2025-11-20
> **Easier comments addressed first; rest is coming as soon as experiments finish**
>
> Thanks for your suggestions and comments. We address them below.
>
> ### W1 (presentation)
> Thanks for this feedback. We replaced some narrative citations with parenthetical ones to improve readability. We replaced interrupting inline citations with narrative ones, to avoid interrupting the flow of the text, especially in the introduction. Lastly, what do you mean with “applied ICLR format”?
>
> ### W2 (incorrect claim 2; reformulate claim 1)
> Thank you for pointing this out. We revised claim 1 for precision and clarified claim 2 by explicitly showing how it follows from our gradient-based optimization framework. The updated text should make these contributions unambiguous.
>
> We modified claim 1 from: *“Unlike previous methods that only remove edges, our framework permits both edge additions and node feature modifications. With the sole exception of extra self-loop additions, this allows us to perturb the entirety of the original input data that the oracle processes.”* to: *“Our framework supports edge additions, edge removals, and node feature perturbations, enabling perturbation of nearly all input degrees of freedom. This richer perturbation space allows us to explore how structural and attribute-level changes jointly influence the oracle’s prediction process and to uncover richer, truly counterfactual motifs that edge-only methods cannot express.”*
>
> Claim 2 from: *“By exploiting the properties of the loss, we find the closest counterfactual through directed modifications. This guarantees that the counterfactual explanation is not only minimal but also consistent with the oracle’s learned decision boundaries.”* to: *“By exploiting the properties of the loss, we find counterfactuals that are locally closest in the loss-optimization landscape, as guided by directed gradient-based modifications (see Appendix A.1). This guarantees that the counterfactual explanation is not only minimal under the objective but also consistent with the oracle’s learned decision boundaries.”*
>
> ### W6 (on the trade-off between validity and distance)
> We agree that counterfactual validity must be considered jointly with distance. In our formulation, this trade-off is governed by the beta term in Eq. 2, which plays the same role as in prior work. As for the trade-off between validity and distance to the original graph, we show GED, sparsity, and CS to the original instance. We refer to that in Sec.3.1:  $L_{pred}$ is a prediction loss that encourages $\Phi(G^\prime)\ne \Phi(G)$ and $L_{dist}$ is a distance loss that promotes similarity between $G^\prime$ and $G$, with the trade-off controlled by $\beta$. We refer to K (the total number of iterations of the algorithm) in Appendix C.2; we have now provided a more precise specification of this. We also transparently show validities in the main tables and the GED-CS relationship in Fig. 3b.
>
> ### Q1 (claim 1 suggest XPlore deletes nodes; CLEAR already does that)
> Our method does not perform node deletion. It jointly performs edge deletion, edge insertion and node feature perturbation. CLEAR doesn’t do node deletion (see [1] and the Appendix D of [2]). CLEAR learns a latent distribution through VAEs and then samples from there. When sampling, they approximate graph matching, which doesn't consider node deletions. So CLEAR can either add or delete edges. It cannot operate at the node level. To see this in action, you can also check GRETEL [3] to run a quick code snippet and verify that CLEAR doesn’t do node deletions.
>
> ### Q2 (claim 2 and problems with closest counterfactual)
> Thank you for pointing this out. We clarified claim 2, which was not expressed correctly (see W2), by explicitly showing how it follows from our gradient-based optimization framework and referencing material covered in Appendix A.1, which explains convergence to local minimizers. The updated text should make these contributions unambiguous. Laslty, however, Table 3 doesn't show GEDs; rather in Fig. 3b we show the relationship between GED and CS scores.
>
> ### Q3 (your method does diffusion; compare better against D4Explainer)
> We don’t do diffusion in this paper. We’re unsure why the reviewer is requesting this comparison. See Q.7.
>
> ### Q4 (Eq. 2 is weirdly introduced here)
> Eq. 2 introduces the full counterfactual search objective, while Eq. 7 defines only its prediction component $L_{pred}$​. The purpose of Eq. 2 is to clearly state the global optimization target before detailing its constituent terms. We adopted this ordering for consistency with prior work by Lucic et al., which presents the loss and its components in the same structure.
>
> [1] Prado-Romero et al. A survey on graph counterfactual explanations: definitions, methods, evaluation, and research challenges. CSUR'23
>
> [2] Prado-Romero et al. Robust stochastic graph generator for counterfactual explanations. AAAI'24
>
> [3] Prado-Romero et al. Developing and evaluating graph counterfactual explanation with GRETEL. WSDM'23

---

> ### Author Response · Authors · 2025-11-20
> **Second part of easier comments**
>
> ### Q5 (issues with node explanations)
> Node counterfactual explanations have been described in the work of Lucic et al. The dataset now consists of instances of the form $(v, X_v, A_v)$, where $v$ is a target node, and $(X_v, A_v)$ is the subgraph induced on $v$. The oracle $\Phi$ must be a classifier that takes in input instances $( X_v, A_v)$ and predicts the label of $v$. Our Eq. 3 must change to $\mathcal{E}(X_v,A_v,P)=\text{softmax}(\bar{D_v}^{-½}(P \odot A_v +I)\bar{D_v}^{-½} X_vW)$.  Now, finding a counterfactual means changing Eq. 2 into $L(v,v’) = L_{pred}(v,v’|\Phi,\mathcal{E}) + \beta L_{dist}(v,v’)$ ,where $L_{pred}(v,v’|\Phi, \mathcal{E}) = -\mathbf{1}[\Phi(X_v,A_v) = \Phi(X_{v’},A_{v’})] \cdot L_{logits}(\Phi_{\ell-1}(X_v,A_v),\Phi_{\ell-1}(X_{v’},A_{v’}))$. The distance function can be anything as long as it measures it node-wise. For example, our Eq. 8 would have a $v$ in the suffix. We can put these details in the appendix. However, we don't believe that including them in the main paper would provide the reader with any new information, since Lucic et al. have already introduced node counterfactuals.
>
> ### Q6 (Eq. 3 is a subcase of Eq 4. in Lucic et al.)
> Comparing our Eq. 3 to Eq. 4 in Lucic et al., both formulations utilize an edge-masking operator. However, Lucic et al. perform counterfactual search on a localized subgraph around the target node, while our formulation operates on the entire graph adjacency, leading to a strictly larger and different search space. Therefore, Eq. 3 is not a restricted instance of their model, but rather a distinct, graph-level formulation (or more generic formulation if you wish). We take inspiration from Lucic et al’s work, but framing our Eq. 3 as a subcase of their Eq. 4 would, in our opinion, be misleading. Furthermore, Eq. 4 is the final formulation by Lucic et al., while our Eq. 3 is one of the steps used to derive our final formulation, Eq. 6.
>
> ### Q7 (reframe XPlore as noising-denoising)
> Our perturbations are not generated through a noisy-denoising process. We only add stochastic noise at initialization to enhance exploration and regularization, a standard practice in optimization [1-3]. Since XPlore does not involve a forward diffusion process or a denoising sampler, framing it as a diffusion process would be misleading in this context.
>
> [1] Mo Zhou et al., Towards Understanding the Importance of Noise in Training Neural Networks. ICML'19.
>
> [2] Samuel L. Smith et al., On the Generalization Benefit of Noise in Stochastic Gradient Descent. ICML'20.
>
> [3] Noh et al., Regularizing Deep Neural Networks by Noise: Its Interpretation and Optimization. NIPS'17.
>
> ### Q8 (Tab. 3 is misleading; where is GED?)
> We would like to point you to our analysis of all these aspects. Table 3 presents validity and fidelity, and to complete the trade-off analysis, we show GED and CS in Figure 3.b and runtimes in Fig. 4 in the appendix. In Table 8 in the appendix, we also provide details on sparsity and the number of oracle calls for each dataset.
>
> ### Q9 (XPlore stops only if a valid counterfactual is found $\rightarrow$ unfair comparisons)
> If no counterfactuals are found, XPlore stops after K=50 iteration steps, which is one-tenth of the K=500 steps needed by Lucic et al. Neither method stops at the first counterfactual; instead, remaining iterations are used to find closer solutions according to the objective. Hyperparameter search has been performed and discussed (see Appendix C.2). Even with a limited budget, Xplore consistently achieves high validity, demonstrating the effectiveness of our perturbation strategy. Baselines were run with their default budgets (as specified by the authors), and analysis of XPlore’s strong performance is presented in Sec. 4.2. Lastly, we invite you to notice that XPlore can fail to find counterfactuals (see AIDS, MUTAG, BBBP, among others in Tab. 3), and it simply stops the procedure. So, could you please explain to us what you mean by *"[...] getting 100 % validity is not surprising [...]"*?
>
> ### Q11 (Sec. 4.3)
> Sec. 4.3 examines how OOD regions of the feature space impact counterfactual generation, which is directly tied to our third stated objective of robustness and is considered in the design of our method. This section highlights that, although the OOD effect is partly mitigated via cosine similarity, it still persists and has not been completely eliminated, informing the reader about the method’s limitations and motivating future work on more robust oracle models. We will provide a clearer reference to this section and present it more effectively.
>
> ### Q12 (Proof in A.1)
> We will streamline this. We thought of providing it for others to follow the entire discussion here, without having to jump from paper to paper, just to understand what’s going on. But we agree that this can be eliminated.

---

> ### Author Response · Authors · 2025-11-21
> **Literature Review is outdated: check COMBINEX, Verma et al. and Kosan et al.**
>
> ### W3, W4, W5 (outdated related works; you stop at 2023; see Verma et al. 2023, Kosan et al. 2023, and Giorgi et al. 2025)
> We included, to the best of our knowledge, the most recent one, which is D4Explainer (NeurIPS 2023). We are also aware of GIST (ICML 2025) [1], but at the time of writing, it was only on arXiv, and we considered it as contemporary work. Kosan’s paper, which you mention, is for model-level explanations, not instance-level explanations, which is our scenario. This means that Kosan’s method provides a single explanation for the entire dataset or model, rather than single counterfactuals per instance based on the model’s predictions. We will make this distinction clearer in the related work.
>
> Verma et al. focus exclusively on node-level explanations, whereas our work also produces graph-level explanations. Their method perturbs only the topology, through edge additions, without performing feature perturbations, which our approach includes. A key difference is that Verma et al. train a dedicated explainer model on top of the oracle, whereas our method does not require training any explainer. Their framework is inductive, meaning the explainer is trained once and then applied to any node in the graph, in contrast to methods that train one explainer per node. Our method, while training-free, is also general: the same algorithm can be applied to any node or graph and optimizes explanations directly, which is arguably a more flexible and principled strategy. We'll include Verma's paper in the related work.
>
> We will include COMBINEX in the related works. We conducted initial experiments to compare XPlore with COMBINEX on the same datasets used by both (AIDS, PROTEINS, and Enzymes). We ran COMBINEX with the same oracle as we did for the rest of the SoTA, using the same dataset splits. We used the dynamic variant of COMBINEX with the hyperparameters specified in the paper (i.e., lr = 0.1 and 500 epochs with Adam optimizer). XPlore outperforms COMBINEX across the board. Unfortunately, we were unable to reproduce the same results as in the original paper. We’re currently running COMBINEX on the other datasets and will report the results as soon as they are available.
>
> |  | AIDS |  | PROTEINS |  | ENZYMES |  |
> |---|---|---|---|---|---|---|
> |  | Validity $\uparrow$ | Fidelity $\uparrow$ | Validity | Fidelity | Validity | Fidelity |
> | COMBINEX | 0.015 | 0.015 | 0.027 | 0.027 | 0.100 | 0.083 |
> | XPlore | **0.323** | **0.323** | **0.654** | **0.627** | **1.000** | **0.942** |
>
> Weirdly, COMBINEX uses the distribution distance described as the L2 distance between the embedding of the counterfactual and the mean of the embeddings of the dataset. We argue that this is an artifact and an incorrect metric, as it doesn’t verify the similarity between the original instance and the produced counterfactual. Taking the mean of the dataset doesn’t make sense; it would be more reasonable if this mean were taken on the counterfactual class, and then one would measure the distance of the counterfactual from this class-aware mean. So, we ran the GED on both to ensure a fair comparison. Here’s the table of GEDs:
>
> |  | AIDS | PROTEINS | ENZYMES |
> |---|---|---|---|
> | COMBINEX | 5.02 | 12.55 | 121.83 |
> | XPlore | 216.84 | 80.16 | 277.78 |
>
> It is evident that COMBINEX does fewer perturbations than XPlore; however, its lower validity undermines its utility. We argue that having higher validity is more important, as, according to Eq. 1, it is a hard constraint in the learning objective (see  the $\arg \min$ condition).
>
> We’ll integrate this method into our current Table 3 (and report details of GED and CS scores accordingly, and fill the full tables in the appendix.
>
> [1] Prenkaj et al. Graph Inverse Style Transfer for Counterfactual Explainability. ICML'25

---

> ### Author Response · Authors · 2025-11-25
> **On the relationship between GED and CS**
>
> ### Q10 unsurprisingly, the GED/CS of the proposed method is much higher than that of other methods, since the algorithm.
>
> We apologize, but we don’t understand what the reviewer means here. We believe there may be a misunderstanding: the GED/CS scores reported for XPlore are comparable to those of existing methods across datasets (cf. Fig.3 b). XPlore does not systematically produce higher GED/CS, and its behaviour is in line with competing explainers (ranking in the top-3 CS on all datasets except COX2). Notably, XPlore achieves consistently higher CS than CF-GNNExpl despite identifying more (i.e., harder) counterfactuals, highlighting its stronger semantic quality.
>
> We show additional figures in our anonymous GitHub (see `additional_figures.md`). The violin-distribution plot of CS and GED (Add-Fig. 0 and Add-Fig. 1) shows that XPlore captures both easy and hard counterfactuals, resulting in a bimodal distribution of CS and GED values. By identifying harder counterfactuals, XPlore may have higher average CS and GED scores, so reporting only the mean can be misleading. For instance, on BAS XPlore has a lower mean CS than iRand or RSGG-CE, yet it performs better on easy samples and achieves roughly double their accuracy, highlighting its effectiveness beyond simple averages.

---

### Official Review · Reviewer_awWW · 2025-10-28

**Soundness:** 2
**Presentation:** 3
**Contribution:** 2
**Rating:** 4
**Confidence:** 2

**Summary:**

This paper proposed a gradient-based framework for graph counterfactual explanations that expands the search space beyond edge deletions to allow edge insertions and node feature perturbations. It optimizes a soft objective balancing prediction flip and distance to the original graph, and naturally extends to node-level counterfactuals.

**Strengths:**

1.	The proposed method XPlore achieves impressive improvements on both validity and fidelity metrics.
2.	The method’s performance was validated on 14 datasets, spanning multiple graph types.
3.	This paper explicitly acknowledged that the residual OOD effects remain open and links them to robustness of oracles.

**Weaknesses:**

1.	To my current knowledge, there exists prior work (e.g., C2Explainer) that has already enabled edge insertion and node feature perturbations; this might weaken Xplore’s claimed novelty unless positioned more precisely.
2.	While the evaluation was performed on 14 datasets, it is skewed towards molecular/biology category, with only one social network dataset (i.e., COLLAB). As social network analysis might be a key application area for GNN interpretability, adding more datasets in this area would benefit generality.
3.	A few typos: (i) in Figure 2 (d), the caption reads “edge inserion” and should be “edge insertion”; (ii) in Section 4.2, “sparisity” should be “sparsity”.

**Questions:**

1.	Could you please situate the proposed work’s novelty against recent counterfactual explainers that support edge insertions and/or node feature perturbations, such as C2Explainer?
2.	Would you consider including recent baselines (2024-2025) that permit node perturbations or edge insertions?
3.	Would you consider adding more social-network datasets beyond COLLAB? This would help assess generality.

---

> ### Author Response · Authors · 2025-11-20
> **First iteration of the rebuttal**
>
> We thank the reviewer for the time and careful consideration given to our work. We have revised the manuscript to address the valuable suggestions provided.
>
> ### W1 and Q1 (C2Explainer)
> Can you provide us with a reference to this method?
>
> ### W3 (typos)
> Thanks, we fixed those typos.
>
> ### Q2 (recent baselines)
> Could you please provide us with references? We included, to the best of our knowledge, the most recent one, which is D4Explainer (NeurIPS 2023). We are also aware of GIST (ICML 2025) [1], but at the time of writing, it was only on arXiv, and we considered it as contemporary work.
>
> ### W2 and Q3 (more social network datasets)
> Since we tested Xplore on 14 datasets spanning five different domains (i.e., synthetic, molecular, bioinformatics, computer vision and social network), we believe that our method is already sufficiently generic (as also acknowledged by you in Strength #2). We are adding more social network datasets and we will report results as soon as they are ready in this conversation.
>
> [1] Prenkaj et al. Graph Inverse Style Transfer for Counterfactual Explainability. ICML’25.

---

> ### Author Response · Authors · 2025-11-29
> **W2 and Q3 experiments**
>
> We're still waiting on TWITTER to finish since it's huge and our computational capacities are limited. However, below you can find the results for other datasets as requested. **bold** is best; *italic* is second. DBLP, IMDB and TWITTER are social-network datasets; MSRC is a computer vision dataset.
>
> |  | DBLP  |  |  IMDB | | MSRC  | |
> |---|---|---|---|---|---|---|
> |  | Validity $\uparrow$ | Fidelity $\uparrow$ | Validity | Fidelity | Validity | Fidelity |
> | CF2 | 0.058 | 0.027 | 0.506 | 0.366 | 0.909 | 0.371 |
> | CLEAR | 0.007 | 0.005 | 0.562 | 0.420 | 0.240 | 0.012 |
> | D4Explainer | 0.070 | 0.031 | 0.496 | 0.358 | 0.909 | 0.371 |
> | iRand | 0.012 | 0.006 | 0.037 | 0.026 | *0.957* | 0.391 |
> | RSGG-CE | *0.575* | *0.450* | **0.861** | **0.664** | **1.000** | **0.423** |
> | CF-GNNExpl | 0.058 | 0.027 | 0.506 | 0.366 | 0.909 | *0.371* |
> | XPlore| **0.918** | **0.748** | *0.788* | *0.606* | **1.000** | **0.423** |

---

> ### Author Response · Authors · 2025-12-01
> **We found C2Explainer and here's what's different between us and them**
>
> We believe that XPlore is a SoTA contribution that goes beyond C2Explainers according to the following dimensions. We will cite C2Explainer [1] and discuss it in our related works.
>
> 1. **Unspecified Supergraph Constraint**
>
> C2Explainer restricts edge additions to an initial "supergraph" that pre-selects all admissible candidate edges. Edges outside this set can never be added, even if they would be the most effective for altering the prediction. However, the paper provides no explanation (conceptual, mathematical, or code-level) of how this supergraph should be computed. Users are simply told they may supply any mask they prefer, leaving the process entirely heuristic and unguided. XPlore does not rely on this opaque constraint.
>
> 2. **Over-Simplified Node Feature Perturbation**
>
> C2Explainer perturbs node features only through binary gating, which severely limits the expressiveness of the explanations. In contrast, XPlore supports free and continuous updates of node features (*XPlore w/ freedom*).
>
> 3. **Similarity Metric Ignores Changed Node Features**
>
> C2Explainer uses a similarity term equivalent to our sparsity, ignoring node features entirely. As a result, it can drastically alter node features while still reporting a high "semantic similarity," producing explanations that may be structurally close but semantically meaningless. XPlore’s cosine similarity (CS) incorporates both edges and node features, yielding a substantially more faithful and diagnostic similarity measure.
>
> 4. **Binary Logit Discretization Harms Gradient Quality**
>
> C2Explainer discretizes logits into binary targets inside the loss. This collapses informative gradients and makes all edge updates identical in magnitude, leading to coarse, unstable optimization. XPlore preserves the real-valued logits, producing a much smoother loss landscape and more precise, targeted updates.
>
> 5. **Narrow and Synthetic-Heavy Empirical Evaluation**
>
> C2Explainer evaluates graph explanations on only two datasets (one synthetic, one molecular) and node explanations on four datasets, all synthetic. This yields a highly unbalanced and limited empirical view. XPlore instead conducts a comprehensive and domain-balanced evaluation: 18 datasets for graph explanation and 5 for node explanation (before the discussion period, there were 14 for graph explanation and 1 for node), covering molecular, social, synthetic, and heterogeneous domains.
>
> [1] MA, Jiali; TAKIGAWA, Ichigaku; YAMAMOTO, Akihiro. C2Explainer: Customizable Mask-based Counterfactual Explanation for Graph Neural Networks. In: Proceedings of the 2025 ACM Conference on Fairness, Accountability, and Transparency. 2025. S. 137-149.

---

### Official Review · Reviewer_CW7Q · 2025-11-01

**Soundness:** 3
**Presentation:** 2
**Contribution:** 2
**Rating:** 6
**Confidence:** 5

**Summary:**

The paper studies counterfactual generation on graphs by allowing not only edge deletions but also edge additions and feature perturbations. The proposed gradient-based framework optimizes these operations in a unified manner. It replaces distance-based objectives with a cosine-similarity metric, yielding more coherent explanations. Experiments report substantially higher validity and fidelity than state-of-the-art baselines.

**Strengths:**

- Clear algorithmic description with well-structured steps; the method is easy to follow.
- Extensive experiments with comparisons against multiple competing approaches.
- Thoughtful discussion of future directions that can guide subsequent research.

**Weaknesses:**

- The contributions are repetitive and could be more concise. For example, contribution points 1 and 4 appear overlapping and could be merged.
- Positioning the work as an “extension” of a prior paper weakens the novelty message.
- Novelty is limited in parts; for instance, edge addition in counterfactual explanations has prior art.
- The motivation for feature perturbations is underdeveloped, and the ablation on this component is limited.
- The search space and resulting computational complexity are not sufficiently justified.
- It remains unclear why the method should yield better out-of-distribution robustness or influence.

**Questions:**

- Edge additions for counterfactual explanations have been studied. What is the specific new insight or advantage your approach provides over prior formulations?
- Complexity: With edge operations, a naive search could appear O(n^2). Please clarify why your algorithm remains O(|E| + n f)?
- Motivation for feature perturbation: Could you add concrete examples where edge edits alone fail but small feature changes produce plausible, faithful counterfactuals (e.g., molecular graphs where atom attributes change properties, or social/product graphs where node attributes shift recommendations)?
- Experimental protocol: How many runs were executed for the explanation module? You report standard deviations—does the table show mean ± std over k runs? Please state k and any fixed random seeds.

Editorial/Presentation Notes:
- Figure 2 colors are hard to distinguish; maybe increase line/marker thickness for clarity.

---

> ### Author Response · Authors · 2025-11-19
> **Part 1 of the rebuttal**
>
> We thank the reviewer for the time and careful consideration given to our work. We have revised the manuscript to address the valuable suggestions provided. We will divide this response into two comments to distinguish between weaknesses (**W**s) and questions (**Q**s).
>
> ### **W1**: more concise contributions.
> Thank you for your suggestion; we have updated our contributions accordingly.
>
> ### **W2, W3, Q1**: novelty and advantages of XPlore vs. SoTA.
> While edge additions have appeared in prior work, existing methods typically (i) rely on heuristic rules or generative models, and (ii) don't treat node-feature perturbations. In contrast, XPlore provides a unified, gradient-based formulation that jointly optimizes edge additions, edge deletions, and continuous node-feature modifications under a single objective. This leads to a significantly more flexible search space while remaining computationally simple (see **Q2**).
>
> A second advantage is interpretability of the explainer itself. Under the assumption of having access to the gradients of the oracle, XPlore remains transparent unlike generative approaches such as CLEAR, RSGG-CE, or D4Explainer, which introduce an additional learned (and black-box) model. This avoids the need to explain a second black-box model and keeps the explanation pipeline lightweight and faithful to the underlying decision boundary.
>
> We will clarify this in the main text and make our contributions clearer as per **W1**.
>
> ### **W4**: feature perturbations and ablations studies on their effect.
> We already provided ablations on feature perturbations in Sec. 4.4 (Tables 5 and 6). Feature perturbations are a core component of our method and strongly contribute to its performance; therefore, evaluating their impact is a natural part of our study. Table 5 reports validity across datasets, showing that all three variants consistently outperform CF-GNNExpl, with the best results achieved when node features are perturbed. Table 6 analyzes how these variants behave as gamma varies. We will make this contribution clearer in the main text and in the experiments.
>
> ### **W5**: search space and computational complexity.
> Including feature perturbations enlarges the search space, but the computational cost remains modest because our explainer does not train an additional black-box model. It performs a single gradient-based optimization over continuous relaxations of structure and features (e.g., treating them as probabilities), which scales linearly with the number of optimization steps and avoids combinatorial enumeration. Empirically, this results in oracle calls and runtimes comparable to those of CF-GNNExpl, despite the broader space (see Table 8 and Figure 4), as gradients guide the search directly. We would also like to point out that, in Appendix A.7, we analyze the computational complexity of an iteration step of Algorithm 1. We’ll make links from the main text to the appendix clearer to support the reader.
>
> ### **W6**: OOD robustness.
> Our claim is not that XPlore eliminates OOD effects, but that it mitigates them relative to methods operating at the same perturbation level. In Fig. 3(b), when we compare explainers at similar GED values, XPlore consistently achieves higher CS between the original and counterfactual graph embeddings. Since CS reflects semantic alignment, higher CS indicates that XPlore’s counterfactuals remain closer to the original graph’s latent semantics rather than drifting into OOD regions. Competing methods with comparable GED often obtain much lower CS, showing that their counterfactuals diverge semantically and thus rely more heavily on OOD artifacts. XPlore’s joint optimization over edge and feature perturbations allows it to make smaller, semantically coherent adjustments, which explains its improved robustness under similar edit budgets. Thus, XPlore handles OOD cases better on the same GED scale because it preserves semantic structure more effectively, even though, as discussed in Sec. 4.3, OOD effects are not fully removed.
>
> Unfortunately, in some cases, exploiting OOD instances to generate “fictitious counterfactuals” is incentivized by the unstable oracle’s decision boundary. Some works (e.g., Leemann et al. [1]) explore non-adversarial counterfactual explanations in other data types; however, a judge model is needed to verify the non-adversarial nature of the produced counterfactual. In [1], this judge is the ground truth, which isn’t a realistic scenario to have access to the labels. We will explore this in the future.
>
> [1] Leemann T, Pawelczyk M, Prenkaj B, Kasneci G. Towards Non-adversarial Algorithmic Recourse. In World Conference on Explainable Artificial Intelligence 2024 Jul 10 (pp. 395-419). Cham: Springer Nature Switzerland.

---

> ### Author Response · Authors · 2025-11-19
> **Part 2**
>
> ### **Q2**: complexity (why $O(|E| + n\cdot f)$ and not $O(n^2)$?)
> We will clarify this in the main text. A naive search over all possible edge operations would indeed be $O(n^2f)$, where $n$ is the number of nodes and $f$ is the number of node features. However, XPlore does not enumerate edge combinations. Instead, it optimizes continuous edge and feature perturbation matrices via gradient descent. Each optimization step involves:
> * Message passing over edges, which follows the standard GNN cost of $O(|E|\cdot d)$ where $d$ the hidden dimension.
> * Node–feature perturbation updates, applied independently across all nodes and features, costing $O(n\cdot d\cdot f)$.
>
> Therefore, each iteration of XPlore runs in $O(|E|\cdot d + n\cdot d\cdot f)$, which reduces to $O(∣E∣+n\cdot f)$ when treating $d$ as a small constant. If the graph is dense $|E| \sim n^2$, then the complexity reduces to $O(n^2)$ as you noticed. Nevertheless, in practice, graphs are sparse, and if not, one can still use sparse-matrices to memorize them in PyTorch for example. A detailed derivation of the time complexity is provided in Appendix A.7.
>
> ### **Q3**: Could you add concrete examples where edge edits alone fail but small feature changes produce plausible, faithful counterfactuals?
> Thanks for your suggestion. We now illustrate this in Fig. 2, where we have corrected the coloring for clarity. I hope you can see it once we upload the new paper pdf with the differences we made.
>
> ### **Q4**: How many runs were executed for the explanation module?
> We report the mean $\pm$ standard deviation over five runs ($k = 5$) for each dataset.

---

### Official Review · Reviewer_Pu3X · 2025-11-01

**Soundness:** 4
**Presentation:** 3
**Contribution:** 3
**Rating:** 8
**Confidence:** 4

**Summary:**

The paper proposes XPlore, a counterfactual explainer for GNNs that can delete and insert edges and also perturb node features. The authors formulate counterfactual search as minimizing a prediction-change loss plus a distance loss. The method targets both graph-level and node-level counterfactuals and introduces a cosine-similarity-on-embeddings metric to capture semantic fidelity. Across 14 datasets, their method outperforms nearly all baselines, and OOD performance is discussed.

**Strengths:**

- XPlore covers graph modifications that most GCE methods don't. Insertions + feature shifts matter.
- Performance on benchmarks against baselines is very strong.
- The authors are honest about their OOD performance and highlight key challenges for all methods.

**Weaknesses:**

- The method still relies on an oracle model, which adds another degree of freedom for practitioners to consider and means XPlore also likely inherits the oracle's faults.
- The OOD discussion seems to suggest that XPlore focuses on model-flipping counterfactuals rather than plausible counterfactuals, which is a significant weakness.

**Questions:**

- Are there any ablations for examining deletions only, deletions+ insertions, and deletions+insertions+features performance?
- Have the authors examined examples for the molecular datasets to ensure that the generated counterfactuals are also chemically valid (e.g. does not violate valence rules)?
- Table 4 only considers CF-GNNExpl on one dataset. How about other baselines/datasets?

---

> ### Author Response · Authors · 2025-11-20
> **First iteration of the rebuttal**
>
> We sincerely thank the reviewer for their positive assessment and for the confidence reflected in the high score. Your encouraging feedback motivates us to further strengthen the work during the rebuttal phase. In particular, we aim to clearly demonstrate XPlore’s advantages over SoTA graph counterfactual explainers. Below, we address each of the raised concerns in detail. We are currently running additional experiments directly targeted at your comments and will report the corresponding tables as soon as they are complete.
>
> ### W1 (the oracle)
> The oracle (as denominated by Prado-Romero et al. [1]) is simply the prediction model whose output XPlore explains; XPlore is not an additional black-box model. Thus, no extra model needs to be trained. As with any post-hoc explainer, XPlore inherits the oracle’s limitations, but in practice, its explanations can expose such shortcomings (e.g., unstable or uninformative rationales), signaling the need to improve the underlying model.
>
> ### W2 (OOD suggests XPlore does model-flipping)
> Figures 3.a and 3.b do not suggest that XPlore produces implausible counterfactuals. Counterfactuals are produced according to. the decision boundary of the oracle. This means that if the oracle has not been trained to mitigate adversarial (which is our case), then it might happen that OOD graphs get incorrectly classified. In fact, this has also been shown by Leemann et al. [2] where plausibility and non-adversarialness of the produced counterfactuals are two separate concepts. XPlore indeed produces plausible counterfactuals (see Fig. 3.b) where the “semantic” similarity between the original graph and the counterfactual is highest w.r.t. our competitors that report similar GED scores. However, in some cases (see Fig. 3.a) it might happen that the oracle “can get fooled” by OOD data and classify these counterfactuals as valid. As shown in [2], a judge model is needed to verify the non-adversarial nature of the produced counterfactual. In [2], this judge is the ground truth, which isn’t a realistic scenario to have access to the labels. We will explore non-adversarial graph counterfactuals in the future.
>
> ### Q1 (ablations for examining deletions only, deletions+ insertions, and deletions+insertions+features)
> The requested ablations are already reported in Tables 5 and 6. For clarity, we use simplified names; **deletions only** correspond to **CF-GNNExpl**; **deletions+insertions** correspond to the variant **w/o gating and freedom**; and **deletions+insertions+features** correspond to the variants **w/ gating** or **w/ freedom**, where the two differ in whether node features are gated or allowed to change freely. We will make this connection clearer in the main text. Furthermore, Table 5 reports validity across datasets, showing that all three variants consistently outperform CF-GNNExpl, with the best results achieved when node features are perturbed. Table 6 instead analyzes how these variants behave as $\gamma$ (i.e., the value used to populate the $\Gamma$ matrix entries that correspond to missing edges of $A$, see Algorithm 1) varies.
>
> ### Q2 (molecular validity of counterfactuals)
> We do not enforce chemical validity (e.g., valence constraints), as our focus is to analyze the oracle’s decision boundaries rather than to generate synthesizable molecules. Chemical validity could be delegated to the oracle or integrated as an optional domain-specific constraint in future extensions. However, with this paper we wanted to demonstrate XPlore advantages across domains under a strict counterfactual evaluation protocol as reported in [1,3].
>
> ### Q3 (Table 4 has only one dataset)
> We would like to draw your attention to the SoTA methods and their limitations in only performing explanations at the graph level. Therefore, only CF-GNNExpl is suitable for comparison, as shown in the node-level explanations in Table 4. We are running other experiments at node-level explanations and will report experiment results as soon as they are ready.
>
> [1] Prado-Romero et al. A survey on graph counterfactual explanations: definitions, methods, evaluation. CSUR’23
>
> [2] Leeman et al. Towards Non-adversarial Algorithmic Recourse. In World Conference on Explainable Artificial Intelligence 2024
>
> [3] Guidotti. Counterfactual explanations and how to find them: literature review and benchmarking. Data Mining and Knowledge Discovery. 2024.

---

> > ### Author Response · Authors · 2025-11-27
> > **Here are the experiments on node-level explanations (cf. Q3)**
> >
> > |  | BZR |  |  ENZYMES |  | TWITTER  |  |  MSRC  |  |
> > |---|---|---|---|---|---|---|---|---|
> > |  | Validity $\uparrow$ | Fidelity $\uparrow$ | Validity | Fidelity | Validity | Fidelity | Validity | Fidelity |
> > | CF-GNNExpl | 0.072 | 0.071 | 0.122 | 0.001 | 0.265 | 0.000 | 0.072 | 0.000 |
> > | XPlore | **0.999** | **0.998** | **0.811** | **0.695** | **0.997** | **0.034** | **0.488** | **0.419** |

---

> > ### Author Response · Authors · 2025-12-03
> > **OOD visualizations (cf. W2)**
> >
> > We added additional figures on the anonymous GitHub (see `OOD_figures.md`) where it can be seen how XPlore is often able to produce plausible counterfactuals. Sometimes XPlore is the only explainer landing in the correct distribution, this shows how other explainers sometimes do not produce plausible counterfactuals as well. We have updated sec 4.3 accordingly.
> >
> > Results show the t-SNE projections of Wavelet Characteristic embeddings for the Tree-Cycle dataset. They show a propension for XPlore to correctly find counterfactual for the Tree class (CF is Cycle), while CFs for Cycle class are a weakness; and the propension for RSGG to find CFs for the Cycle class (CF is Tree) while struggling to find CFs for the Tree class.
> >
> > Comparison with CFGNNE is also analyzed: XPlore here always behave correctly, while sometimes RSGG struggles to land in-distribution counterfactuals.

---

### Meta-Review · Area_Chair_8wwY · 2025-12-23

**Summary:**

The most positive aspect about the work is the use of a gradient-guided optimization framework which the reviewers viewed as elegant, avoiding the need to train an additional black-box explainer and keeping explanations closely tied to the oracle’s decision boundary.
The breadth of the empirical analysis is also regarded as an additional strenght, at least for what pertains the number of benchmarks considered (the coverage of baseline models is instead an issue). Reviewer Pu3X also appreciated the transparent discussion of out-of-distribution (OOD) behavior, acknowledging remaining challenges rather than overclaiming robustness.

At the same time, reviewers raised several strong concerns.

The most significant weakness is novelty positioning: while the unified formulation is clean, edge insertions and node-feature perturbations have appeared in prior work, and some reviewers felt the contribution is more incremental than initially claimed. Relatedly, parts of the literature review and baseline coverage were initially outdated, omitting several recent counterfactual explainers, as highlighted by the specific references provided by several reviewers in their comments.

Concerns were also raised about plausibility, especially for molecular graphs, since chemical validity constraints are not enforced.

From a presentation standpoint, reviewers noted clarity and conciseness issues, with overlapping contribution statements and dense exposition in parts of the paper. Worringly, reviewers also noted issues with some of the main paper claims.

**Reviewer Concerns:**

Concerns about OOD behavior were handled carefully: the authors clarified that XPlore does not eliminate OOD effects but improves semantic alignment under comparable edit budgets, supported by additional figures and analysis.

Feedback on presentation issues (issue with claim statements, typos, unclear equations) was largely addressed through revisions that improved clarity and organization. Requests for experimental protocol details were answered by specifying that results are averaged over multiple runs with fixed seeds.

New baselines and datasets, including more social-network datasets and comparisons with recent methods, were added during the discussion phase.

In general, the issue about novelty positioning remains. While the difference with respect to prior work is preliminarly discussed in the authors rebuttal, this does not convincingly support the fact that the paper constitutes a significant advancement with respect to the state of the art. In particular, it is evident a lack of empirical comparison with the most recent and related works. Without such a comparison it is not evident if the claimed diversity against releated works is also substantial.

**Reviewer Scores:**

Reviewer Pu3X had a very positive assessment of the work (8), which however was not backed up by a substantial review (in terms of depth and specificity). No issues about the novelty of the work and lack of relevant literature was raised. I am under the impression that such aspect could have been noted by the reviewer during the discussion with the other reviewers and might have led to a reduction of the score.

Reviewer CW7Q was marginally positive with high confidence: rebuttal from authors was mostly on point, aside the novelty aspects. I could expect this to result in the reviewer maintaining the score.

Reviewer awWW was marginally negative with limited confidence, and again issues about novelty aspect. Also this reviewer may have stayed with the current assessment.

Reviewer mmHe had substantial criticisms about the overall quality of the work (presentation wise, in terms of technical issues with the claims) as well as in terms of novelty. I expect this reviewer to have stayed with their current score given the unconvincing addressing of the issues as regards missing recent baselines models in the empirical analysis.

Overall this projects to an expected average score ranging in 4.5-5.0, which is below the acceptance bar.

---

### Decision · Program_Chairs · 2026-01-26

Reject